# Efficient Object-Centric Representation Learning using Masked Generative Modeling

**Akihiro Nakano**                                    *nakano.akihiro@weblab.t.u-tokyo.ac.jp*
*Graduate School of Engineering*
*The University of Tokyo*

**Masahiro Suzuki**                                    *masa@weblab.t.u-tokyo.ac.jp*
*Graduate School of Engineering*
*The University of Tokyo*

**Yutaka Matsuo**                                    *matsuo@weblab.t.u-tokyo.ac.jp*
*Graduate School of Engineering*
*The University of Tokyo*

**Reviewed on OpenReview:** *https://openreview.net/forum?id=t9KvOYPeL3*

## Abstract

Learning object-centric representations from visual inputs in an unsupervised manner has drawn focus to solve more complex tasks, such as reasoning and reinforcement learning. However, current state-of-the-art methods, relying on autoregressive transformers or diffusion models to generate scenes from object-centric representations, suffer from computational inefficiency due to their sequential or iterative nature. This computational bottleneck limits their practical application and hinders scaling to more complex downstream tasks. To overcome this, we propose MOGENT, an efficient object-centric learning framework based on masked generative modeling. MOGENT conditions a masked bidirectional transformer on learned object slots and employs a parallel iterative decoding scheme to generate scenes, enabling efficient compositional generation. Experiments show that MOGENT significantly improves computational efficiency, accelerating the generation process by up to 67x and 17x compared to autoregressive models and diffusion-based models, respectively. Importantly, the efficiency is attained while maintaining strong or competitive performance on object segmentation and compositional generation tasks.

## 1 Introduction

A key aspect of human intelligence is the ability to perceive their surroundings as composition of objects and their relationships (Spelke, 1990; 2013). Such abstraction allows humans to flexibly generalize and reason about novel scenarios by composing existing conceptual knowledge, a capability known as compositional generalization in machine learning (Greff et al., 2020; Goyal & Bengio, 2022; Lake et al., 2017). Inspired by this ability, the field of object-centric learning aims to develop models that can decompose complex scenes, such as images or videos, into individual object representations in an unsupervised manner. A prevalent approach in object-centric learning is to represent each object in an image or video as a set of representations, often referred to as "slots" (Burgess et al., 2019; Greff et al., 2019; Locatello et al., 2020). Early works achieved object discovery and disentanglement by introducing various inductive biases, such as grouping nearby pixels (Greff et al., 2017; van Steenkiste et al., 2018), modeling object properties (e.g. position, size, depth, etc.) explicitly (Eslami et al., 2016; Jiang et al., 2020), and modeling foreground and background separately (Lin et al., 2020b;a).

Among various architectures, Slot Attention (Locatello et al., 2020) emerged as an influential method, employing iterative attention (Vaswani et al., 2017) over encoded features to bind information into distinct

object slots and reconstructing the scene using a mixture-based decoder (Watters et al., 2019). Trained on a simple input reconstruction objective, Slot Attention is a popular architectural choice and has been extended for both images (Singh et al., 2022a; Seitzer et al., 2023; Didolkar et al., 2025) and videos (Singh et al., 2022b; Kipf et al., 2022; Zadaianchuk et al., 2023; Wu et al., 2023b). Recent works have focused on enhancing the generative capabilities of these slot-based approaches. These models often leverage the extracted slots as conditional inputs for sequential generators such as autoregressive transformers or diffusion models such as Latent Diffusion Model (LDM) (Rombach et al., 2022), enabling object-centric disentanglement and compositional generation on more realistic datasets (Singh et al., 2022a;b; Wu et al., 2023b; Jiang et al., 2023; Kakogeorgiou et al., 2024).

However, these models often suffer from computational inefficiency. For example, when using an autoregressive transformer, generation requires (# of patches) steps per image (Figure 1 (a)). This is especially challenging in object-centric learning, as effective object-centric disentanglement typically requires smaller patches to capture objects of varying sizes and partial occlusions accurately. On the other hand, diffusion-based models significantly reduce the number of generation steps required per image, yet remain computationally expensive memory-wise and time-wise due to their iterative refinement procedure (Wu et al., 2023b; Jiang et al., 2023). Though the broader field of generative modeling has actively pursued solutions to improve efficiency—such as using inherently efficient models (Chang et al., 2022a; Song et al., 2023; Wang et al., 2025), optimizing sampling (Song et al., 2021; Ni et al., 2024), and applying post-training acceleration techniques (Luhman & Luhman, 2021; Luo et al., 2023)—this focus has not yet carried over to object-centric learning. Our work is the first to bridge this gap. We address this problem at the architectural level as it not only provides a model that is inherently efficient while preserving the generative quality but also establishes a basis for post-hoc acceleration methods.

In this work, we present MOGENT (**M**asked **O**bject-centric **GEN**erative **T**ransformer), an object-centric learning framework that leverages the efficiency of parallel decoding using masked generative modeling. MOGENT employs a masked bidirectional transformer decoder, conditioned on extracted slots, to predict masked visual tokens representing image patches. While inspired by the success of parallel decoding schemes on image and video generation (Chang et al., 2022a; Yu et al., 2023; 2024), successfully integrating this paradigm with object-centric representation learning is a non-trivial challenge. A naive integration fails because the unstructured, parallel nature of the decoder is not inherently suited for object-centric learning which requires spatial locality priors (Chakravarthy et al., 2023). We empirically show that integrating the Query Slot Attention (QSA) (Jia et al., 2023) to extract slots and adjusting the initialization and loss function of the transformer is crucial for achieving effective object-centric representation learning within this efficient framework. MOGENT can generate a large number of tokens in parallel at each step, significantly improving computational efficiency. For example, generating a 128x128 image requires only 20 decoding steps with MOGENT, a substantial reduction from the 1024 steps typically required by autoregressive baselines (Singh et al., 2022a). More importantly, MOGENT can generate efficiently regardless of the image resolution as the number of steps to generate does not depend on the image resolution. Experiments on 3D Shapes (Burgess et al., 2019) and CLEVR (Johnson et al., 2017) datasets show that MOGENT achieves highly efficient inference speed compared to relevant baselines. Furthermore, experiments on CLEVRTex (Karazija et al., 2021) and CelebA (Liu et al., 2015) datasets show that our model is scalable to more realistic datasets in a highly efficient manner, up to 67x and 17x speedup compared to autoregressive and diffusion-based models, respectively. Notably, this efficiency gain is realized without compromising, and often improving upon, representation learning and generation quality.

## 2 Related Works

**Object-centric Learning.**    Learning to represent objects in the scene using "slots" (Locatello et al., 2020; Burgess et al., 2019; Greff et al., 2019) has been long explored in the literature. A key inductive bias to achieve object-centric disentanglement is iterative inference, which has been achieved by applying iteration over objects (Eslami et al., 2016; Burgess et al., 2019) or iterative refinement (Greff et al., 2019; Locatello et al., 2020; Goyal et al., 2021). Additionally, adding further priors about object properties (e.g. position, size, depth, etc.) (Eslami et al., 2016; Jiang et al., 2020), foreground and background (Lin et al., 2020a;b), or temporal indifference (Hsieh et al., 2018; Nakano et al., 2023), has also been found to be effective in improving object-centric disentanglement. Slot Attention (Locatello et al., 2020) is one of the commonly-used model,

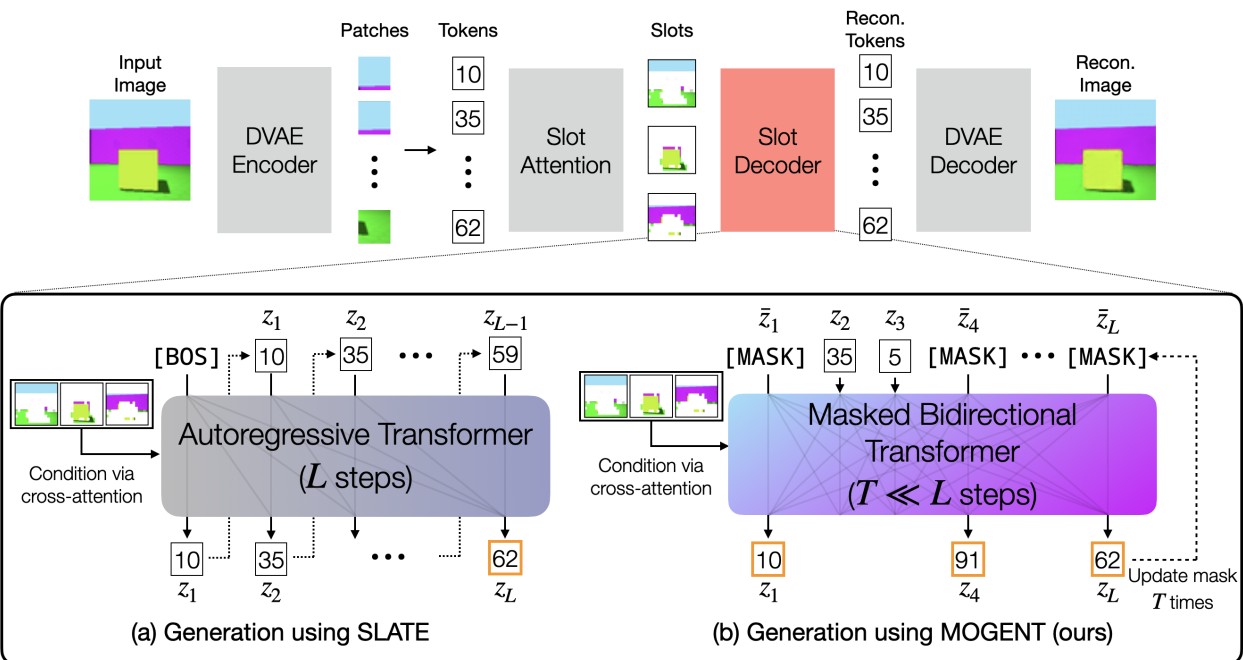

Figure 1: Overview of the generation process using (a) SLATE and (b) MOGENT. As SLATE uses an autoregressive transformer for slot-to-token decoding, generation takes as many steps as the number of tokens ($L$) to represent a single image. On the other hand, by employing a masked bidirectional transformer, MOGENT can generate tokens in parallel, reducing the number of steps to generate by a large margin.

which uses iterative attention mechanism (Vaswani et al., 2017) and mixture-based decoder (Watters et al., 2019) to learn slot representations from various datasets. However, the efficacy of these early object-centric models is often limited when dealing with complex real-world scenes (Yang & Yang, 2022). To address this, several works have explored improving Slot Attention, such as employing bi-level optimization (Chang et al., 2022b; Jia et al., 2023), adding spatial locality prior (Chakravarthy et al., 2023), or learning quantized slot representations (Singh et al., 2023; Wu et al., 2024).

Other works have explored improving generation performance of slot-based models by replacing the mixture-based decoder with models with higher capacity, such as transformer or diffusion models (Singh et al., 2022a;b; Sajjadi et al., 2022; Wu et al., 2023a;b; Jiang et al., 2023; Kakogeorgiou et al., 2024). For example, SLATE (Singh et al., 2022a) and STEVE (Singh et al., 2022b) use an autoregressive transformer to generate images or videos from slots, respectively. They train a discretized VAE (Im et al., 2017) to tokenize the inputs, extract slots using Slot Attention, and generate scenes using a slot-conditioned transformer decoder. In contrast, SlotDiffusion (Wu et al., 2023b) employs LDM (Rombach et al., 2022) to generate scenes using slot-conditioned denoising within the latent space of a pretrained VQ-VAE (van den Oord et al., 2017). While powerful, both the sequential token prediction in autoregressive models and the iterative nature of diffusion lead to computational inefficiency and slow generation times (Wu et al., 2023b; Jiang et al., 2023). Addressing this bottleneck, our work employs masked generative modeling, aiming for more efficient object-centric learning and parallelizable generation.

**Accelerating Generative Models.** Recently, improving the efficiency of image generation methods is gathering attention as it is a crucial factor for real-world utilization. Advancements for both autoregressive-based and diffusion-based models have been explored. To accelerate autoregressive models, most works have worked on using non-autoregressive approaches. For example, MaskGIT (Chang et al., 2022a) and its following works utilizes masked token prediction and bidirectional Transformer for efficient generation with iterative sampling scheme (Chang et al., 2023; Yu et al., 2023; 2024). Other approaches explore a combination of autoregressive and non-autoregressive methods (Wang et al., 2025; Li et al., 2024c). Compared to the autoregressive counterpart, acceleration of diffusion models have been studied more in depth, including new architectures such as consistency models (Song et al., 2023; Luo et al., 2023) and using improved sampling

schemes via faster samplers (Song et al., 2021; Lu et al., 2022; 2025). We tackle the efficiency problem using non-autoregressive models, as we empirically show that it gives a better point on the speed-quality curve compared to other models. For a broader taxonomy of acceleration strategies, see Appendix D.

**Masked Generative Modeling.** The probelm of slow inference speed and sequential error accumulation by autoregressive decoding have been extensively studied in the field of natural language processing. Non-autoregresive generation, especially masked token prediction, emerged to address this challenge (Devlin et al., 2019; Ghazvininejad et al., 2019; Mansimov et al., 2019). Application to images has also been explored (Chang et al., 2022a; Lee et al., 2022), in which MaskGIT (Chang et al., 2022a) improved both image generation quality and efficiency. MaskGIT has been extended to video prediction (Yan et al., 2023), text-to-image (Chang et al., 2023; Bai et al., 2025), text-to-video (Yu et al., 2023; 2024; Villegas et al., 2023), multi-modal generation (Chang et al., 2023; Kim et al., 2023; Mizrahi et al., 2024), LiDAR point generation (Zhang et al., 2018), motion generation (Guo et al., 2024; Pinyoanuntapong et al., 2024a;b), and neural simulation of interactive environments (Bruce et al., 2024). Our work is the first to investigate and adapt masked generative modeling to object-centric representation learning.

## 3 Method

We begin by reviewing the background of object-centric learning using SLATE (Singh et al., 2022a) (Section 3.1), which we build our model on. We then detail our object-centric masked generative transformer architecture and explain how to perform compositional image generation using the learned representations of MOGENT. (Section 3.2). Figure 1 illustrates the architecture of SLATE and MOGENT.

### 3.1 Preliminary: Slot-based object-centric learning using SLATE

The goal of object-centric learning is to learn a set of representations, or "slots", that each correspond to an object within a scene. A commonly-used architecture is Slot Attention (Locatello et al., 2020), which learns slot representations by computing iterative attention between randomly initialized slots and encoded input image. SLATE (Singh et al., 2022a) extends this work by combining Discrete VAE (DVAE) (Im et al., 2017) and an autoregressive transformer decoder (Vaswani et al., 2017).

Specifically, SLATE encodes an input image $\mathbf{x}$ through the DVAE encoder $f_\phi$, to produce log probabilities, $\mathbf{o}$, for a categorical distribution with $V$ classes. A "soft" one-hot encoding $\mathbf{z}_{\text{soft}}$ is sampled from a relaxed categorical distribution (Jang et al., 2017), and decoded via the DVAE decoder, $g_\theta$. Denoting the temperature of the relaxed categorical distribution as $\tau_{\text{DVAE}}$, the entire process can be written as

$$\tilde{\mathbf{x}} = g_\theta(\mathbf{z}_{\text{soft}}) \text{ where } \mathbf{z}_{\text{soft}} \sim \text{RelaxedCategorical}(\mathbf{o}; \tau_{\text{DVAE}}), \ \ \mathbf{o} = f_\phi(\mathbf{x}). \tag{1}$$

To compute slots, the tokens from the DVAE encoder are first mapped to embeddings, $\mathbf{e}$, using a learned dictionary. Learned positional embeddings, $\mathbf{p}_\phi$, are added to the embeddings to incorporate positional information of the tokens. Then, the embeddings are fed to Slot Attention (Locatello et al., 2020) encoder to extract $K$ slots, $\mathbf{s}_{1:K}$. This process can be written as,

$$\mathbf{s}_{1:K} = \text{SlotAttention}(\mathbf{e}) \text{ where } \mathbf{e} = \text{Dictionary}_\phi(\mathbf{z}) + \mathbf{p}_\phi, \ \ \mathbf{z} \sim \text{Categorical}(\mathbf{o}). \tag{2}$$

Finally, starting from a `[BOS]` token, an autoregressive transformer (Vaswani et al., 2017), $p_\theta$, decodes the slots back into the discrete tokens one at a time, which can be formulated as generation using next token prediction:

$$p_\theta(z_1, \cdots, z_L | \mathbf{s}_{1:K}) = \prod_{l=1}^{L} p_\theta(z_l | z_1, \cdots, z_{l-1}, \mathbf{s}_{1:K}), \tag{3}$$

where $L$ denotes the number of tokens. The resulting tokens can be decoded back into an image by the DVAE decoder, $g_\theta$, enabling compositional scene generation.

Overall, DVAE is trained to minimize the negative log-likelihood, $\mathcal{L}_{\text{DVAE}} = \mathbb{E}_{\mathbf{z}_{\text{soft}}}[-\log g_\theta(\mathbf{x}|\mathbf{z}_{\text{soft}})]$, using reconstruction loss. Slot Attention and the transformer decoder are trained to minimize the negative log-likelihood, $\mathcal{L}_{\text{ST}} = \mathbb{E}_{\mathbf{s}_{1:K}}[-\sum_{l=1}^{L} \log p_\theta(z_l | z_1, \cdots, z_{l-1}, \mathbf{s}_{1:K})]$, using cross-entropy loss. The entire model is trained together. Please refer to Singh et al. (2022a) for more information on training details.

## 3.2 MOGENT

As explained in the previous section, most object-centric generative models generate new scenes by first inferring the slot representations and then decoding them back to the pixel space. While the choice of the slot-to-token decoder is important for both effective object-centric disentanglement and high generation quality (Wu et al., 2023a), existing options present notable trade-offs. Mixture-based decoders (Watters et al., 2019) are computationally efficient but often possess limited capacity, tending to produce blurry results on complex data (Singh et al., 2022a; Wu et al., 2023a). In contrast, decoders with higher capacity such as autoregressive transformer-based (Singh et al., 2022a;b; Kakogeorgiou et al., 2024) or latent diffusion model-based decoders (Wu et al., 2023b; Jiang et al., 2023) generate higher-quality images but are computationally expensive, hindering training and inference speed..

To address these limitations, we propose MOGENT, a framework leveraging masked generative modeling for object-centric representation learning and efficient compositional generation. Drawing inspiration from the success of masked generative modeling in generating high-quality images and videos with high efficiency (Chang et al., 2022a; Yu et al., 2023; 2024), we utilize the MaskGIT (Chang et al., 2022a) framework as the decoder. Specifically, we view the slot-to-token decoding problem as generation using masked token prediction by replacing the autoregressive transformer of SLATE with BERT (Devlin et al., 2019), a transformer with bidirectional attention.

During training, the bidirectional transformer is trained to predict the masked parts of the input tokens. A binary mask, $\mathbf{m}(r) = [m_l]_{l=1}^{L}$, is generated using a predefined masking scheduler function, $\gamma(r) \in (0, 1]$ as follows: first sample a ratio, $r$, from a uniform distribution, $\mathcal{U}(0, 1)$, then uniformly select $\lceil \gamma(r) \cdot L \rceil$ tokens to mask out of $L$ total tokens. Following MaskGIT, we choose cosine function as the masking scheduler. The token, $z_l$, is replaced with a `[MASK]` token if $m_l = 1$, otherwise unmasked. Denoting the masked input $\bar{\mathbf{z}}^r = \mathbf{z} \odot \mathbf{m}(r)$, we train the bidirectional transformer, $p_\theta$ to minimize the negative log-likelihood of the masked tokens using cross-entropy loss:

$$\mathcal{L}_{\mathrm{ST}} = \mathbb{E}_{\mathbf{s}_{1:K}} \left[ \mathbb{E}_{\mathbf{m}(r) \sim p_{\mathcal{U}}} \left[ -\sum_{l=1}^{L} \log p_\theta(z_l | \bar{\mathbf{z}}^r, \mathbf{s}_{1:K}) \right] \right]. \tag{4}$$

Similar to SLATE, we incorporate cross-attention layers in the bidirectional transformer for slot conditioning.

For inference, we use the iterative parallel decoding scheme of MaskGIT. We start with a blank canvas with all tokens masked out and operate the following procedures iteratively for $T$ steps; (1) Predict the probabilities for all the masked tokens at step $t$, $\bar{\mathbf{z}}^{<t} = \mathbf{z} \odot \mathbf{m}^t$. (2) Sample a token based on the predicted probabilities. (3) Compute the number of tokens to mask using the mask scheduler function. (4) Decide tokens to unmask for the next iteration, $\bar{\mathbf{z}}^t$ using the schedule from (3) and the log probabilities from (1) used as "confidence" score. As $\gamma(r) = \gamma(t/T)$ is a monotonically decreasing function, the iterative decoding scheme ensures that the number of unmasked tokens monotonically increase until all tokens are generated at step $T$. Crucially, at each step, we condition the generation on the slots by leveraging the cross-attention layers to generate scenes as a combination of objects.

Formally, the iterative decoding scheme can be viewed as a generation using "next set-of-tokens prediction" (Li et al., 2024c). Let $\mathcal{S}$ be an ordered list expressing the schedule of unmasking by the scheduler function, $\mathcal{S} = [\bar{\mathbf{z}}^1, \bar{\mathbf{z}}^2, \cdots, \bar{\mathbf{z}}^T]$. Note that $\{\bar{\mathbf{z}}^t\}_{t \in \{1, \cdots, T\}}$ do not contain any overlapping tokens and is complete. Then, the generation using the bidirectional transformer can be expressed as,

$$p_\theta(z_1, \cdots, z_L | \mathbf{s}_{1:K}) = \prod_{t=1}^{T} p(\bar{\mathbf{z}}^t | \bar{\mathbf{z}}^{<t}, \mathbf{s}_{1:K}) = p(\bar{\mathbf{z}}^1 | \mathbf{s}_{1:K}) p(\bar{\mathbf{z}}^2 | \bar{\mathbf{z}}^1, \mathbf{s}_{1:K}) p(\bar{\mathbf{z}}^3 | \bar{\mathbf{z}}^1, \bar{\mathbf{z}}^2, \mathbf{s}_{1:K}) \cdots p(\bar{\mathbf{z}}^T | \bar{\mathbf{z}}^{<T}, \mathbf{s}_{1:K}). \tag{5}$$

Therefore, the generation requires $T$ steps in total, which is non-dependent of the number of tokens, $L$.

Empirically, we find that naively replacing the original slot decoder of SLATE with a masked bidirectional transformer is insufficient for learning object-centric representations. We hypothesize this stems from differences in how spatial locality priors—the assumption that nearby pixels often belong to the same object (Chakravarthy et al., 2023)—are handled. While SLATE's sequential autoregressive decoding naturally focuses on local neighborhoods, the bidirectional attention in MOGENT allows attending to further away

Table 1: Comparison of computation requirements of SLATE and MOGENT using 3D Shapes dataset. All metrics were computed on a single NVIDIA Tesla V100 GPU, with batch size of 64 for training and 1 for test.

|  |  | SLATE | MOGENT (Ours) |
|---|---|---|---|
| Train | # of parameters | 3.6M | 3.7M |
|  | Time [s] | 0.465 | **0.056** |
| Test | Time [s] | 1.929 | **0.182** |

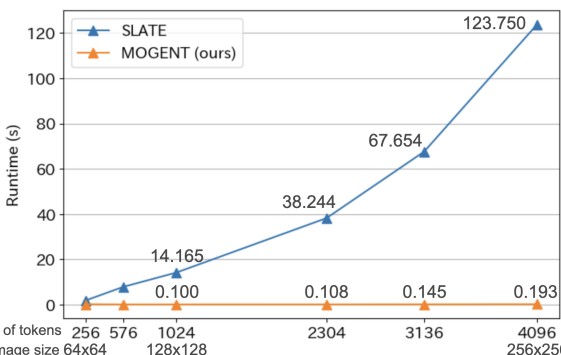

Figure 2: Runtime comparison of image generation between SLATE and MOGENT. All results were computed on a single NVIDIA Tesla V100 GPU.

tokens during decoding, promoting a more global attention but potentially weakening this implicit spatial locality bias.

To mitigate this, we make the following three changes to the model architecture and training setup. First, we adopt Query Slot Attention (QSA) which uses learnable query initializations instead of random initialization. As shown by Chang et al. (2022b); Jia et al. (2023), using random initialization plays a minimal role and can be removed. We empirically find that using QSA leads to large improvements in object-centric disentanglement for our framework. Secondly, we initialize the mask embeddings as 0 (*zero mask init*). This simple technique facilitates the model's ability to differentiate between the slots conditioning and the masked tokens, especially during early training, contributing to more stable learning and better disentanglement. Thirdly, while most previous works on masked generative modeling implement the loss function as the cross-entropy loss on both masked and unmasked tokens, we train MOGENT using cross-entropy loss on only the masked tokens, as derived in Equation 4. This objective discourages the model from learning an identity map on the unmasked tokens, and encourages it to use information from surrounding unmasked tokens to predict the masked tokens. We reason that this loss formulation not only prevents codebook collapse, but also motivates the model to capture the semantic information in the image for better disentanglement. We summarize our empirical findings regarding the model architecture in Section 4.5.

**Compositional Generation.**   The learned slots each represent the individual objects in the image. Therefore, following Singh et al. (2022a); Wu et al. (2023b), we can build a library of the representations from the extracted slots. Then, we can generate images with novel combinations of objects by composing the representations ("concepts") from the library via the following steps: (1) Collect slots from all training images. (2) Apply $K$-means clustering to find $K$ concepts using cosine similarity as the distance metric. (3) To generate a new image, pick concepts from the library and randomly select a slot per concept, and decode using MOGENT and DVAE decoder. Implementation-wise, SlotDiffusion (Wu et al., 2023b)[1] proposes a simplified version of the evaluation in which they generate new images by randomly shuffling the extracted slots within a batch. As they report that the FID result is close to the aforementioned method, we use their implementation to evaluate the performance on this task.

## 4 Experiments

We evaluate the benefits of MOGENT over using an autoregressive transformer decoder in terms of (1) computational efficiency, (2) image segmentation ability, and (3) compositional generation ability, and (4) scalability to more realistic data. We select SLATE (Singh et al., 2022a) as the baseline, as it uses the same transformer-based decoder but trained on next token prediction. For (4), we also compare MOGENT against diffusion-based models, namely SlotDiffusion (Wu et al., 2023b). We evaluate on four datasets with distinct characteristics: 3D Shapes dataset (Burgess & Kim, 2018), CLEVR dataset (Johnson et al., 2017), CLEVRTex dataset(Karazija et al., 2021), and CelebA dataset (Liu et al., 2015). 3D Shapes dataset consists of 400K training images of 3D objects procedurally generated from 6 ground truth independent latent factors, such as

---

[1]https://github.com/Wuziyi616/SlotDiffusion.

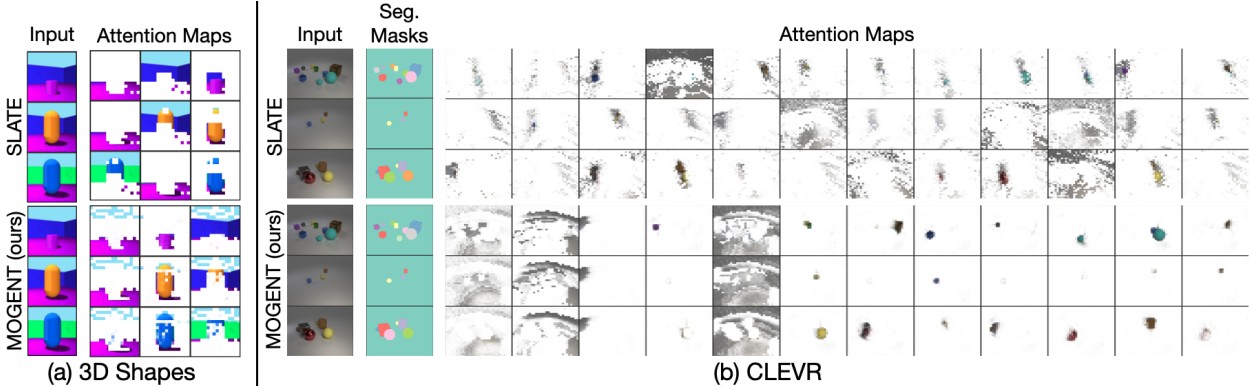

Figure 3: Visualization of attention maps of SLATE and MOGENT on (a) 3D Shapes and (b) CLEVR with masks dataset. For CLEVR, we plot the ground-truth segmentation masks in the second column.

Table 2: Comparison of SLATE and MOGENT on the image segmentation task on CLEVR with masks dataset. We report FG-ARI, mIoU, FG-mIoU, and mBO (mean ± standard deviation over 3 runs).

| | | | | |
|---|---|---|---|---|
| SLATE | 0.611±0.064 | 0.282±0.119 | 0.261±0.105 | 0.267±0.106 |
| MOGENT (Ours) | **0.750±0.116** | **0.466±0.124** | **0.470±0.131** | **0.495±0.125** |

color, size, and shape. CLEVR dataset consists of 200K images of multiple objects with random colors and shapes under photorealistic lighting conditions. CLEVRTex dataset consists of 40K images with a similar setup to CLEVR, but augmented with more diverse object shapes, materials, and textures, and background textures. CelebA dataset consists of 200K images of real-world celebrities with varying background and lighting conditions. We resize the images to $64 \times 64$ for 3D Shapes and $128 \times 128$ for CLEVR, CLEVRTex, and CelebA. We set the number of iteration steps for decoding to $T = 20$ except for its ablational study in Section 4.5. Hyperparameters and training details are summarized in Appendix A.1. Experiment setups for image segmentation and compositional editing tasks are summarized in Appendix A.2.

## 4.1 Computation Efficiency

We evaluated the computational efficiency of MOGENT against SLATE, focusing on training and inference speed. We report the number of parameters, memory consumption, time per training step, and the time required to generate a single image (Table 1). All metrics were measured on a single NVIDIA Tesla V100 GPU. As the table shows, MOGENT has marginally more parameters compared to SLATE, primarily due to an additional linear layer used to project the outputs of the masked transformer decoder into token probabilities. However, MOGENT speeds up training and generation speed around 10 times faster by leveraging the parallel decoding scheme enabled by the masked bidirectional transformer. MOGENT also uses less memory compared to SLATE.

To further investigate the efficiency advantage, we compared the image generation runtime of MOGENT and SLATE across varying image resolutions. As illustrated in Figure 2, the relative speedup offered by MOGENT becomes even more pronounced as image resolution increases.

It is important to note, however, that while MOGENT offers substantial advantages in per-step training and generation speed, it requires more than 2 times of epochs to train compared to SLATE due to its slower convergence. We provide a comparison of their training and validation curves in Appendix B.5.

## 4.2 Image Segmentation

We evaluate how well the models disentangle the images into individual objects. To assess this, we evaluate the image segmentation performance using the CLEVR dataset with ground-truth segmentation masks provided by Greff et al. (2019). We report four metrics; (1) foreground Adjusted Rand Index (FG-ARI), (2) mean Intersection over Union (mIoU), (3) foreground mIoU (FG-mIoU), and (4) mean Best Overlap (mBO).

Table 3: Comparison of SLATE and MOGENT on the compositional generation task measured by FID score and IS (mean ± standard deviation over 3 runs). We report reproduced results for SLATE.

| Dataset | FID (↓) | | IS (↑) | |
|---|---|---|---|---|
| | SLATE | MOGENT (Ours) | SLATE | MOGENT (Ours) |
| 3D Shapes | 49.07±5.06 | **47.27±2.26** | 3.43±0.12 | **3.73±0.15** |
| CLEVR | 100.00±23.29 | **96.51±21.03** | **2.74±0.01** | 2.43±0.03 |

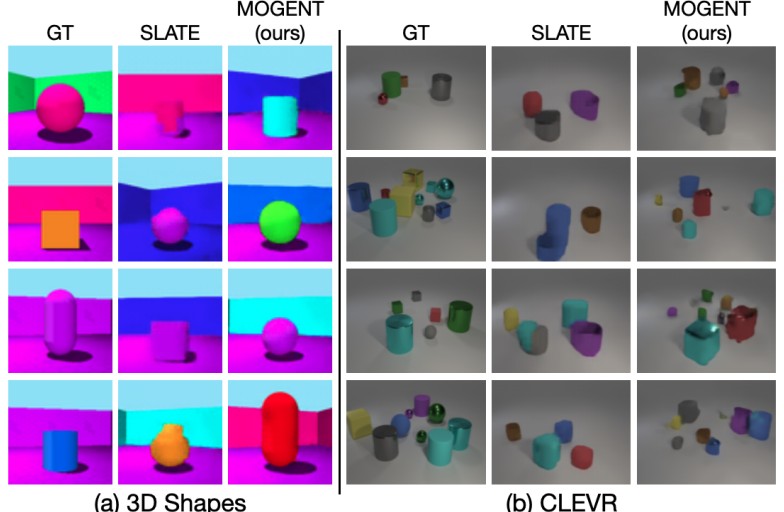

Figure 4: Visualization of compositional generations results on (a) 3D Shapes and (b) CLEVR dataset. Across both datasets, MOGENT is able to generate more realistic and diverse images compared to SLATE.

Figure 5: Visualization of compositional editing on the CLEVR dataset. Red squares represent the target object we aim to swap or remove and red arrows represent with which object we swap the target object with.

These metrics quantify how well the predicted object masks match the ground-truth segmentation. Table 2 presents the segmentation ability of the models. As the table shows, MOGENT outperforms SLATE across all metrics. As shown in Figure 3, MOGENT has learned to disentangle the images into objects better than the baseline model across the two datasets.

## 4.3 Compositional Generation

In this section, we evaluate how well the model is able to generate novel scenes by combining the learned representations. We conduct two experiments; (1) compositional generation task described in Section 3.2 and (2) compositional editing task.

**Compositional generation.** We assess the Fréchet Inception Distance (FID) (Heusel et al., 2017) score and Inception Score (IS) (Salimans et al., 2016) on the compositional generation task in Table 3. We calculate FID score and IS between 40K generated images and the ground-truth images. Figure 4 shows examples of the generated images on both datasets. MOGENT achieves competitive results against SLATE, showing an advantage in mean performance.

In terms of IS, our model shows better score on the 3D Shapes dataset and competitive one for the CLEVR dataset. Combined with the qualitative results, it shows that MOGENT is able to reuse the learned representations to generate new scenes.

**Compositional editing.** In Figure 5, we apply MOGENT to image editing using the learned slots. Using the CLEVR dataset, we randomly swap an inferred slot representation between two images. We swap the slot with either slot of the foreground objects or the background to conduct object swapping or removal, respectively. We use the attention map from QSA slot encoder to mask the tokens where the attention values are high. Then, we generate the image, similar to image inpainting task. While autoregressive models require

Table 4: Comparison of computation requirements of SLATE, SlotDiffusion, and MOGENT using CLEVRTex dataset. All metrics were computed on a single NVIDIA A6000 GPU, with batch size of 64 for training and 1 for test.

Table 5: Compositional generation results on CLEVRTex and CelebA datasets measured by FID score.

|  |  | SLATE | SlotDiffusion | MOGENT (Ours) |
|---|---|---|---|---|
| Train | # of parameters | 10M | 137M | 23M |
|  | Memory [GB] | 45.4 | 15 | 42.7 |
|  | Time [s] | **0.8** | 0.82 | 1.63 |
| Test | Time [s] | 25.5 | 6.72 | **0.38** |

|  | FID (↓) | |
|---|---|---|
|  | CLEVRTex | CelebA |
| SLATE | 88.73 | 78.95 |
| SlotDiffusion | **32.07** | **27.72** |
| MOGENT (Ours) | 76.99 | 67.55 |

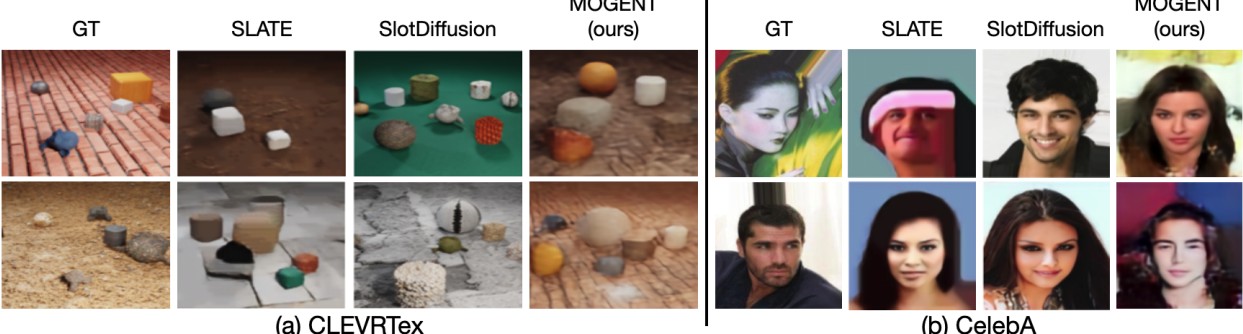

Figure 6: Visualization of compositional generations results on (a) CLEVRTex and (b) CelebA dataset.

the entire image to be generated from scratch, MOGENT can easily edit parts of the image as it does not have any restrictions regarding token prediction orders. As the figure shows, MOGENT is able to swap and insert an object from a different scene. We can also remove objects by swapping the slot with a slot representing the background. Our model generates realistic images that retains the objects' relationships such as appearance and occlusion.

### 4.4 Scaling to More Realistic Data

We further evaluated the scalability of MOGENT to more realistic data, using CLEVRTex and CelebA datasets. We extend our comparison to include SlotDiffusion (Wu et al., 2023b), a diffusion-based generative model, alongside the autoregressive baseline, SLATE. To ensure a fair and relevant comparison, we updated the architecture of both models following the setup in SlotDiffusion. For SLATE, the CNN encoder of SLATE was replaced ResNet18 architecture (He et al., 2016) and the model was trained using a two-step training by pretraining DVAE first and then training the entire model. For MOGENT, we replaced the DVAE with VQVAE (van den Oord et al., 2017), similar to SlotDiffusion, and adopted the two-step training.

**Computation Efficiency.** The computational requirements for all three models on CLEVRTex are compared in Table 4. First, we note the significant differences in model capacity: MOGENT is approximately twice the size of SLATE but contains only 17% of the parameters of the much larger SlotDiffusion model. While MOGENT requires longer training time, its parallel decoding scheme provides a substantial advantage at inference time. MOGENT generates images 67x faster than SLATE and 17x faster than SlotDiffusion. This result underscores our model's primary strength in highly efficient generation, a key bottleneck for autoregressive and diffusion-based approaches. We note that using a transformer-based models require up to 3x more memory. Still, MOGENT requires less memory than the autoregressive baseline, SLATE.

**Compositional Generation.** Table 5 and Figure 6 shows the FID score of the models and examples of generated images, respectively. The table shows that MOGENT improves upon the generative quality of the autoregressive SLATE baseline. However, it does not match the performance of the much larger SlotDiffusion model. We attribute this performance gap in FID score directly to the large difference in model capacity (23M vs. 137M parameters, see Table 4). The primary goal of this paper is to establish the viability and advantages of this efficient paradigm. Thus, our findings firmly establish its value as a highly efficient alternative, while exploring its performance at a larger scale remains a promising direction for future research.

We report further comparison of SlotDiffusion and MOGENT in Appendix B.7.

Table 6: Ablation on using QSA, zero mask init, and calculating loss on only masked tokens (mask loss). We report FID score and IS on the compositional generation task using 3D Shapes dataset.

| Dataset | QSA | zero mask init | mask loss | FG-ARI (↑) | mIoU (↑) | FG-mIoU (↑) | mBO (↑) | FID (↓) | IS (↑) |
|---------|-----|----------------|-----------|------------|----------|-------------|---------|---------|--------|
| 3D Shapes | ✗ | ✗ | ✗ | — | — | — | — | 136.36 | 3.61 |
|  | ✓ | ✗ | ✗ | — | — | — | — | 130.69 | **3.96** |
|  | ✓ | ✓ | ✗ | — | — | — | — | 55.87 | 3.79 |
|  | ✓ | ✓ | ✓ | — | — | — | — | **44.96** | 3.70 |
| CLEVR | ✗ | ✗ | ✗ | 0.299 | 0.258 | 0.181 | 0.219 | 257.29 | 2.40 |
|  | ✓ | ✗ | ✗ | 0.255 | 0.189 | 0.164 | 0.219 | 164.52 | 2.22 |
|  | ✓ | ✓ | ✗ | 0.242 | 0.181 | 0.167 | 0.197 | 198.86 | 1.74 |
|  | ✓ | ✓ | ✓ | **0.747** | **0.462** | **0.468** | **0.489** | **72.23** | **2.46** |

Table 7: Ablation of using RoPE. We report FID score and IS on the compositional generation task for 3D Shapes and CLEVR datasets. We also report segmentation metrics on CLEVR dataset.

| Dataset | RoPE | FG-ARI (↑) | mIoU (↑) | FG-mIoU (↑) | mBO (↑) | FID (↓) | IS (↑) |
|---------|------|------------|----------|-------------|---------|---------|--------|
| 3D Shapes | ✗ | — | — | — | — | **44.96** | **3.70** |
|  | ✓ | — | — | — | — | 52.79 | 3.60 |
| CLEVR | ✗ | 0.747 | 0.462 | 0.468 | 0.489 | 91.35 | 2.33 |
|  | ✓ | **0.852** | **0.576** | **0.581** | **0.595** | **72.23** | **2.46** |

## 4.5 Ablations

**Model Design.**  We conduct an ablation study evaluating three architectural and training design choices of our model explained in Section 3.2 (Table 6). Additionally, we experiment using rotary positional embeddings (RoPE) (Su et al., 2024) in the transformer decoder (Table 7).

As Table 6 shows, adopting QSA (Jia et al., 2023) over standard Slot Attention slightly improves both FID and IS metrics. Moreover, further adding zero mask init on the `[MASK]` token and training the model using cross-entropy loss on only the masked tokens both contribute positively to the model's generation ability, leading to largely improved FID scores. We also visualize extracted slots on the 3D Shapes dataset using t-SNE (van der Maaten & Hinton, 2008) and codebook usage of the transformer decoder in Figure 11 and Figure 12, respectively. As the figure shows, adding zero mask init and mask loss contributes largely in improving model's compositional generation task by enabling better slot disentanglement and avoiding codebook collapse, respectively. These findings show that integrating components in both the slot encoder and decoder is important for achieving effective object-centric representation learning with MOGENT.

Table 7 shows the effect of using RoPE as positional embeddings in the transformer decoder. As the table shows, we see that RoPE is effective particularly when training the model on the CLEVR dataset. We hypothesize that this improvement stems from the characteristics of CLEVR, which features images with more variation in object size, including smaller objects. In such scenarios, adding RoPE allows MOGENT to better utilize the relative distance between tokens to capture local details and distinguish individual objects. Consequently, RoPE reinforces the spatial locality important for effective object-centric learning and helps prevent over-reliance on the global context alone.

**Iteration Number.**  We study the effect of the number of iterations ($T$) on our model by evaluating compositional generation task with different $T$s. As shown in Figure 7, increasing $T$ does not necessarily yield consistent improvements: while higher $T$ leads to improved FID scores, it simultaneously results in decreased IS. This observation differs from previous findings in masked image modeling (Chang et al., 2022a), where both FID and IS initially improved with increasing iterations until a "sweet spot", beyond which performance declined. We hypothesize that, since slots encode both object identity and positional information, it acts as a strong conditioning during generation compared to other conditionings, such as text, label and layout. Therefore, we think that increasing iterations does not enhance the diversity of generated images.

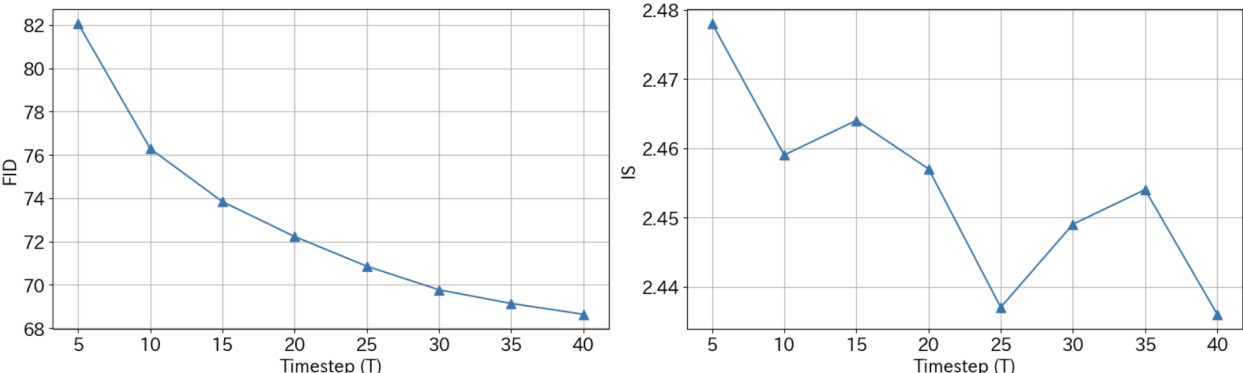

Figure 7: Ablation on the number of iteration steps measured on the compositional generation task using the CLEVR dataset. We report FID score (left) and IS (right).

## 5    Conclusion

In this work, we addressed the computational challenge posed by prevalent object-centric generative models. While previous works offer powerful capabilities for compositional scene generation from object-centric representation, their model architecture makes the model computationally inefficient, particularly during generation. We proposed MOGENT, an object-centric representation learning architecture using a masked generative modeling approach, inspired by MaskGIT. Using the iterative decoding scheme, MOGENT is able to decode slots to tokens in parallel, achieving efficient decoding regardless of input size.

Our results demonstrate that MOGENT reduces computational requirements up to 67x and 17x speedup in generation compared to autoregressive baseline and diffusion-based models, respectively. As the number of steps to generate is independent from the image resolution, the relative speedup by our model increases as the input size increases. Importantly, efficiency is achieved while maintaining the ability to generate images from object-centric representations, achieving better or competitive performance on object segmentation and compositional generation tasks on 3D Shapes and CLEVR datasets. Further experiments on more realistic datasets, namely CLEVRTex and CelebA datasets, show that MOGENT is scalable with improved generation quality compared to the autoregressive baseline. Finally, our ablation studies validate that incorporating appropriate inductive biases, such as using QSA and initializing mask embeddings as zeros, is crucial for effective masked generative modeling and object-centric representation learning. We empirically show that these improvements add the spatial locality bias and avoid codebook collapse of the decoder, both needed for achieving meaningful disentanglement. Overall, our work establishes masked generative modeling as a viable and highly efficient alternative for object-centric generation.

Despite its success, MOGENT has limitations. Firstly, our current work primarily uses synthetic or semi-synthetic datasets where objects are well-defined and separated. In natural scenes, the definition of an "object" becomes more ambiguous; boundaries are often unclear, objects exhibit complex articulation or occlude significantly, and distinguishing foreground objects from the background is non-trivial. Secondly, while significantly faster, the masked generative approach of MOGENT differs from the iterative refinement process of diffusion models. While diffusion models can refine the entire generation throughout their sampling process, masked generative models do not have a mechanism to correct its previously sampled tokens. This lack of a refinement mechanism can potentially lead to uncorrected errors or visual artifacts appearing in the final output (Lezama et al., 2022).

Future work should focus on addressing these limitations. Following prior works (Seitzer et al., 2023; Wu et al., 2023b; Zadaianchuk et al., 2023), further investigation of employing pretrained visual transformers (Dosovitskiy et al., 2021) such as DINO (Caron et al., 2021) or DINOv2 (Oquab et al., 2024), to extract patch-level representations from scenes could yield insights on scaling MOGENT to more complex, real-world image and video datasets. Investigation on hybrid approaches such as masked autoregressive models (Li et al., 2024c) could also help address the inability of masked generative modeling to refine during generation. In addition, we leave the application of MOGENT to downstream tasks such as reinforcement learning and reasoning for future exploration.

**Broader Impact Statement**

Our framework proposes an efficient, object-centric representation learning on various datasets. Our model shows a promising direction towards in using object-centric representations for efficient generation or editing tasks. Since we mainly experiment on synthetic or semi-synthetic datasets, we do not see any immediate risks of human rights violations or security threats in our work. However, future works should investigate the scalability of our work and evaluate the potential risks.

**Acknowledgments**

The authors express special thanks to the anonymous reviewers whose comments led to valuable improvements of this paper. Part of this work is supported by projects commissioned by JSPS KAKENHI Grant Number 24KJ0757, as well as JSPS KAKENHI Grant Number 23H04974.

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

# A   Additional Implementation Details

## A.1   Hyperparameters and Training Details

The hyperparameters used for our experiments are reported in Table 8 and Table 9. We followed the implementation of SLATE (Singh et al., 2022a) and mainly changed only the transformer decoder architecture. Although MaskGIT (Chang et al., 2022a) uses a larger transformer decoder, with 24 layers, 8 attention heads, 768 embedding dimensions and 3072 hidden dimensions, we kept our hyperparameters similar to the transformer decoder used by SLATE to measure performance fairly. The model was trained using Adam optimizer (Kingma, 2015) with $\beta_1 = 0.9, \beta_2 = 0.999$. We used a fixed learning rate of 3e-4 for the DVAE and a learning rate of 1e-4 with linear warmup for stable learning. For training on 3D Shapes dataset, we halved the learning rate if the validation loss did not decrease for 4 consecutive epochs.

Following QSA (Jia et al., 2023), we added a perturbation to the initial slots by sampling from a normal distribution of mean zero and variance $\sigma$ for better performance. We applied cosine annealing to decrease the perturbation from 1 to 0 during training. We set the # of annealing steps to match the # of annealing steps for DVAE's temperature. Following SlotDiffusion (Wu et al., 2023b), we set the # of warmup and annealing steps to match 5% and 15% of the total training steps, respectively. During training, we mask the tokens based on a cosine scheduling: for each training sample, the masking rate is sampled from a truncated arccos distribution with density function, $p(r) = \frac{2}{\pi} \left(1 - r^2\right)^{-\frac{1}{2}}$. This has an expected masking rate of 0.64, showing a bias towards higher masking rate. To train MOGENT, while some works (Wu et al., 2023b; Jiang et al., 2023) that pretraining DVAE leads to better performance, we found that training all components from scratch led to better object-centric disentanglement.

Following SlotDiffusion (Wu et al., 2023b), we make two architectural changes to our model when training on CLEVRTex and CelebA datasets. First, instead of using DVAE encoder to obtain features to extract slots from, we use a ResNet18 (He et al., 2016) encoder. Secondly, we replace the DVAE with VQVAE (van den Oord et al., 2017). The VQVAE is pretrained for 100 epochs with batch size 64 and is kept frozen when training the entire model.

During training, MOGENT requires approximately 45GB of memory, which is equivalent to the requirement of SLATE. Training MOGENT takes around 7 days on a single NVIDIA RTX A6000 GPU, while SLATE is trained in 3 days using the same GPU setup. We find that MOGENT requires around twice the number of training steps for training, as masked token prediction is more difficult than next token prediction. We report # of hyperparameters and training and generation speed in Section 4.1, and show the training and validation loss curves in Appendix B.5.

We reproduced the results for the baseline model, SLATE, as only the code on 3D Shapes dataset was available. To train SLATE, we used the hyperparameters that was reported in the original paper. We trained SLATE for 20 and 40 epochs for 3D Shapes and CLEVR dataset, respectively.

## A.2   Experiment Setup

**Image Segmentation.**   As explained in Section 4.2, we use foreground Adjusted Rand Index (FG-ARI), mean Intersection over Union (mIoU), foreground mIoU (FG-mIoU), and mean Best Overlap (mBO) to evaluate segmentation ability. We use the attention map from the Slot Attention encoder and take the argmax along the slot dimension to obtain the predicted mask. To compute mIoU and FG-mIoU, we use Hungarian matching to obtain the ground-truth and slots assignment. To compute mBO, we assign the ground-truth mask to the slot with the largest overlapping mask, and then averages the IoU of all pairs of masks.

**Compositional Editing.**   We view the compositional editing task as an inpainting task. Given a pair of samples, we first extract slot representations. To swap an object, we identify the slots attending to the foreground objects and randomly swap a slot between the samples. To remove an object, we swap with the slot attending to the background.

Table 8: Hyperparameters of MOGENT.

| Dataset | | 3D Shapes | CLEVR |
|---|---|---|---|
| Batch Size | | 50 | 64 |
| Epochs | | 80 | 400 |
| Learning Rate Warmup Steps | | 30000 | 21860 |
| Max Learning Rate | | 1e-4 | 1e-4 |
| Gradient Clipping | | 1.0 | 1.0 |
| Encoder | Image Size | 64 | 128 |
| | # of Tokens | 256 | 1024 |
| DVAE | Vocabulary Size | 1024 | 4096 |
| | Max Temperature | 1.0 | 1.0 |
| | Min Temperature | 0.1 | 0.1 |
| | Temp. Annealing Steps | 30000 | 65580 |
| | Learning Rate (w/o warmup) | 3e-4 | 3e-4 |
| Slot Attention | # of Slots | 3 | 12 |
| | # of Iterations | 3 | 3 |
| | Slot Dimension | 192 | 192 |
| | MLP Dimension | 192 | 384 |
| | $\sigma$ Annealing Steps | 30000 | 65580 |
| MOGENT | # of Layers | 4 | 8 |
| | # of Heads | 8 | 8 |
| | Embedding Dimension | 192 | 192 |
| | Hidden Dimension | 192 | 192 |

Table 9: Hyperparameters of MOGENT on CLEVRTex and CelebA. Since we mainly adopt the hyperparameters on CLEVR dataset, we only list the different hyperparameters.

| Dataset | | CLEVRTex | CelebA |
|---|---|---|---|
| Encoder | Image Size | 128 | 128 |
| | # of Tokens | 1024 | 1024 |
| | Architecture | ResNet18 | ResNet18 |
| VQVAE | Vocabulary Size | 4096 | 4096 |
| | Vocabulary Dimension | 192 | 192 |
| Slot Attention | # of Slots | 12 | 4 |

To generate the edited image, we first identify the tokens corresponding to the slot that was edited by calculating the overlap between the mask per slot and image region per token. We replace the tokens with `[MASK]` token. Finally, we apply the iterative decoding scheme to generate the edited image.

# B Additional Experiments

## B.1 Image Reconstruction

To assess MOGENT on image reconstruction, we set up by randomly masking the tokens of the input image with a ratio $r$, and use MOGENT's iterative decoding scheme to reconstruct images. We report two metrics, MSE and Learned Perceptual Image Patch Similarity (LPIPS) (Zhang et al., 2018), ( Figure 9). We also provide examples of this process in Figure 8.

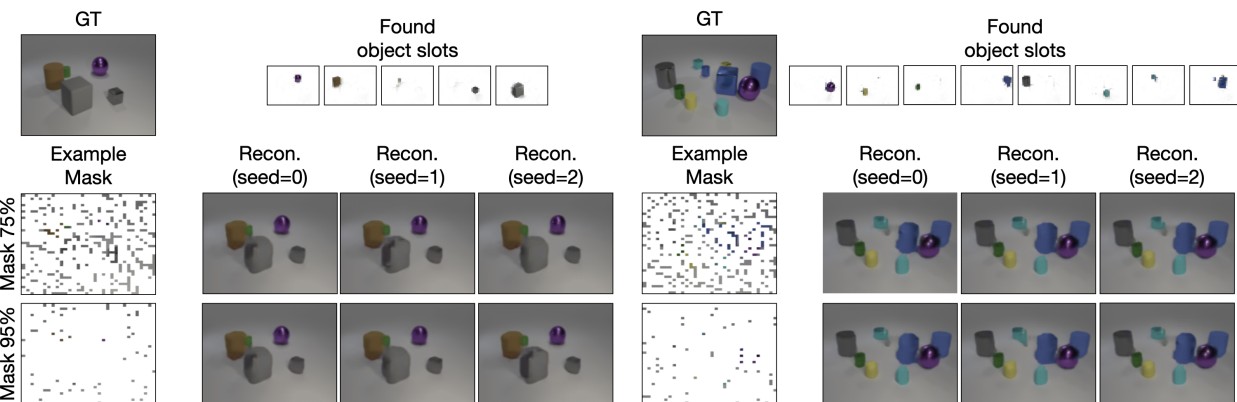

Figure 8: Examples of MOGENT on the image reconstruction task. We show ground-truth image and attention maps of slots for objects in the first row. The second and third row shows example of mask and reconstructed image for $r = 0.75, 0.95$, respectively. On the example on the right, we can see that if MOGENT fails to reconstruct objects if it fails to extract the corresponding slots.

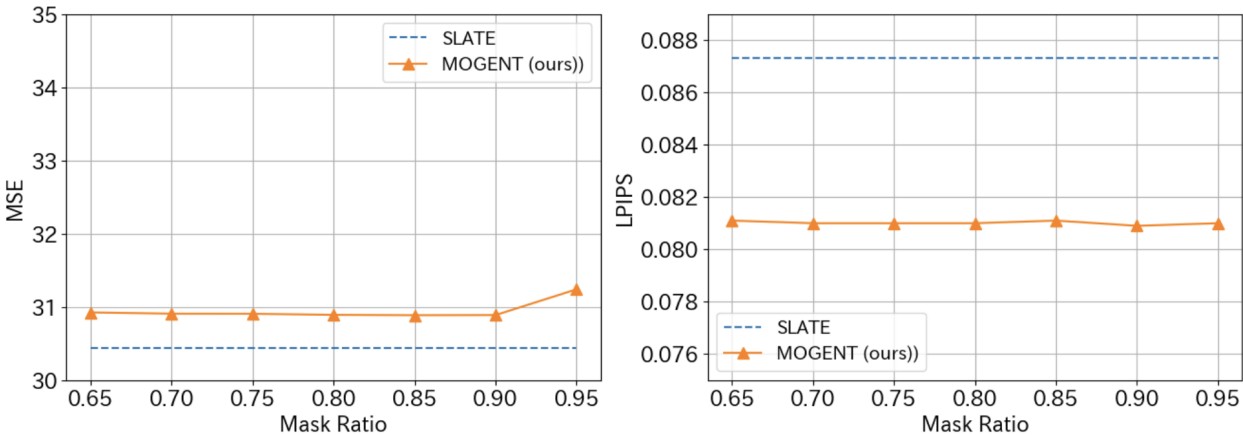

Figure 9: Reconstruction quality vs. mask ratio on CLEVR dataset. We report MSE and LPIPS. We show the score of SLATE in blue as reference.

Interestingly, we find that both MSE and LPIPS do not worsen much even in highly masked setups. Moreover, MOGENT achieves lower LPIPS than SLATE, suggesting that our model reconstructs the images with higher quality than the baseline model. In Figure 8, we compare the reconstruction results between two mask ratio setups. The examples show that even when 95% of the tokens are masked, MOGENT is able to reconstruct the image quite well. We think this is because slot representations contain information about both object identity and position, acting as a strong conditioning about the entire image. In the example on the right, we can see that MOGENT is able to reconstruct all the objects it has found.

## B.2 Analysis on Sampling Temperature

The iterative decoding scheme of MOGENT have an option of adding stochasticity by adding noise to the confidence score. Formally, let $\mathbf{s}_t$ be the confidence score of the tokens at iteration $t$. Then, MOGENT samples the tokens using $\tilde{\mathbf{s}}_t = \mathbf{s}_t + \tau_{\text{TF}} \cdot (t/T)\mathbf{n}$ as the score, where $\mathbf{n}$ is the sampling noise such as i.i.d. Gumbel noise and $\tau_{\text{TF}}$ is the sampling temperature.

We assess the effect of sampling temperature using the compositional generation task on the CLEVR dataset (Figure 10). While MaskGIT used a sampling temperature of $\tau_{\text{TF}} = 4.5$, we found that adding stochasticity to the decoding process leads to degradation in performance. We think this is because our model is conditioned

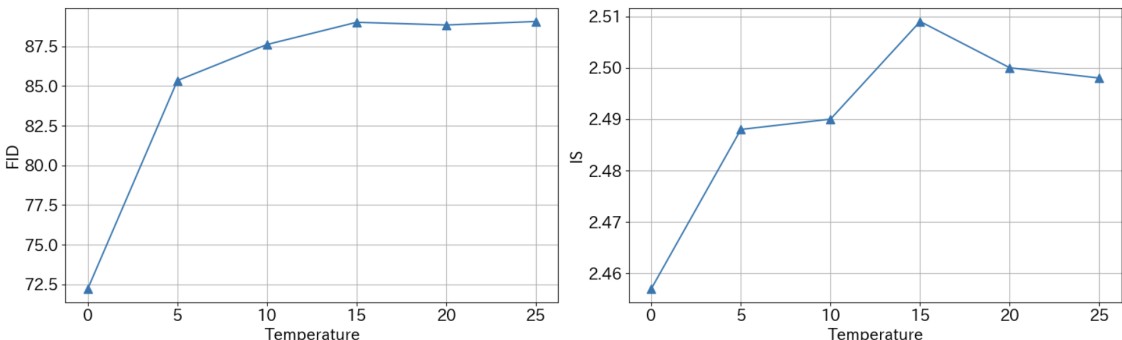

Figure 10: Ablation on the sampling temperature measured on the compositional generation task using the CLEVR dataset. We report FID score (left) and IS (right).

on the slots, which contain information about both object identity and position. As this acts as a strong conditioning on the scene appearance and layout, adding noise during sampling to promote diversity leads to worse generation performance.

### B.3 Failed Attempts

In this section, we provide records of some model variants we experimented.

**Label smoothing.** Following MaskGIT, we experimented applying label smoothing when training the masked transformer decoder. However, we found that this often led to unstable training and worse performance on the compositional generation task.

**Classifier-free guidance (Ho & Salimans, 2022).** We also experimented classifier-free guidance to improve the generation quality of MOGENT. To apply this, we randomly dropped the conditioning (i.e., slots) during training. However, we found that this led to unstable training in which the model did not learn to disentangle the images into individual objects.

### B.4 Further Ablation on Model Design

In this section, we further analyze the performance gain (Table 6) by adding QSA, zero mask init, and mask loss. First, we visualize the extracted slots on the 3D Shapes dataset using t-SNE (van der Maaten & Hinton, 2008). We cluster the slots using $K$-means clustering (Figure 11). As the figure shows, the default model and the model with only QSA have slots that are not disentangled well. By adding zero mask init, we can see that the slots are better disentangled. Secondly, we plot the codebook usage of the transformer decoder (Figure 12). As the figure shows, between the two models with zero mask init, we see an increase in codebook usage by adding mask loss. Thirdly, we visualize the learned attention maps on the CLEVR with masks dataset (Figure 13). From the figure, we can see that mask loss especially improves segmentation by a large margin. These results show that zero mask init and mask loss are especially important as they improve the model's performance by stabilizing training for better slot disentanglement and avoiding codebook collapse, respectively.

Additionally, we experiment using QSA with SLATE and compare the performance gain in Table 10. As the table shows, adding QSA to SLATE yields only a marginal improvement in FID and IS scores. In contrast, MOGENT achieves even better performance, indicating that the majority of the performance gain is attributable to the architectural shift to our masked generative framework.

### B.5 Loss curve

We show the training and validation losses of SLATE, SLATE trained for the same number of epochs as MOGENT (SLATE (extended)), and MOGENT on 3D Shapes dataset in Figure 14. Due to differing loss

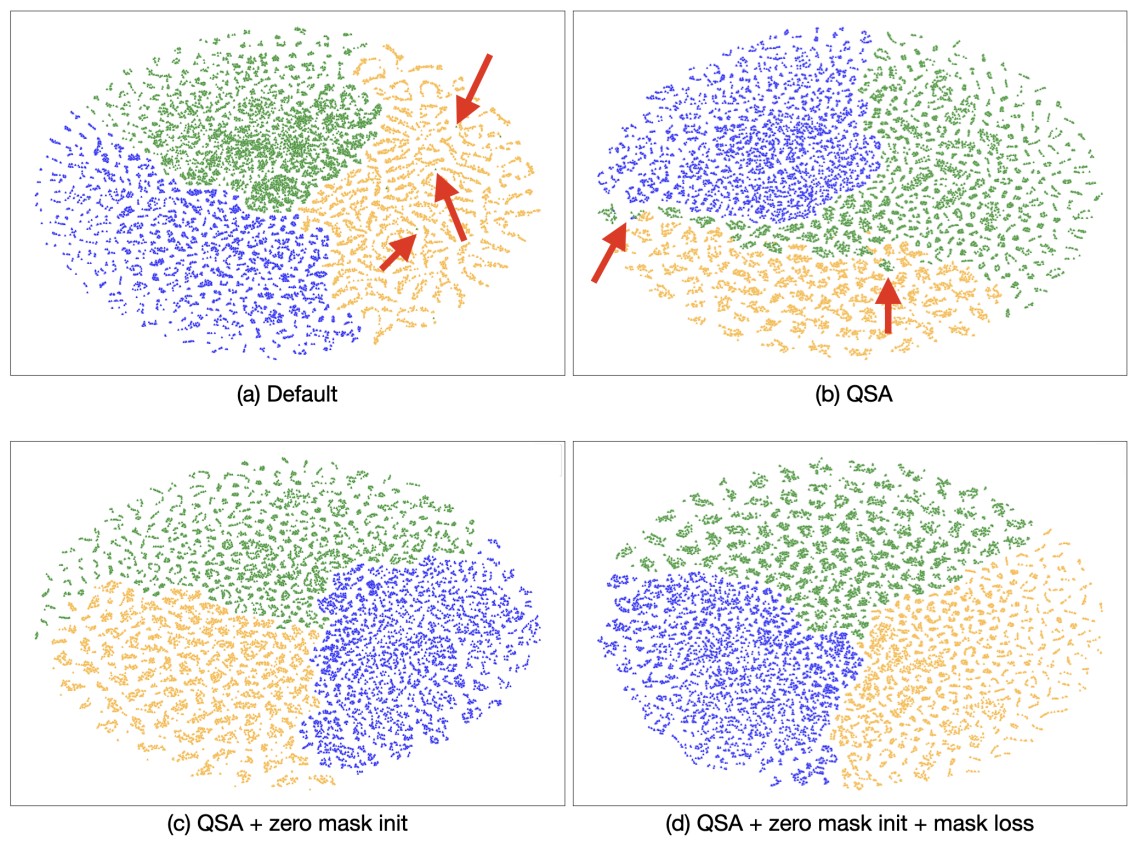

(a) Default

(b) QSA

(c) QSA + zero mask init

(d) QSA + zero mask init + mask loss

Figure 11: t-SNE visualization of extracted slots by different configurations of MOGENT on the 3D Shapes dataset. Colors represent the clustering results using $K$-means clustering. Red arrows show where some slots are not disentangled well.

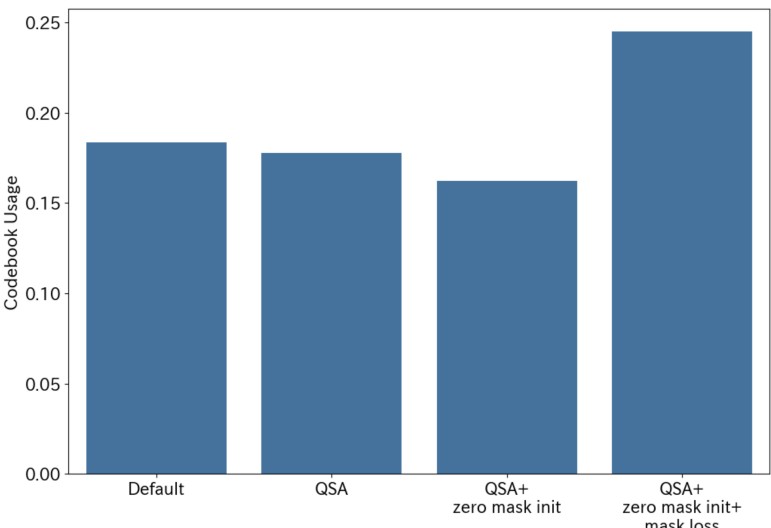

Figure 12: Comparison of the codebook usage of MOGENT's transformer decoder with different configurations.

formulations, the absolute loss values between models are not directly comparable; the figure's primary purpose is to illustrate their respective convergence behaviors.

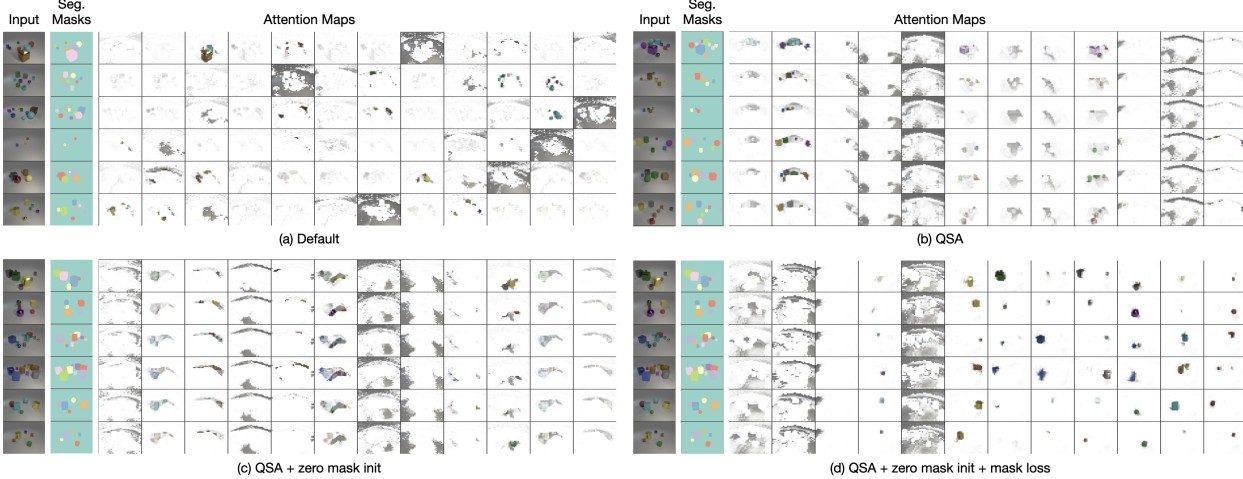

Figure 13: Comparison attention maps by different configurations of MOGENT on CLEVR with masks dataset.

Table 10: Comparison of SLATE, SLATE with QSA (SLATE+QSA), and MOGENT on the compositional generation task using 3D Shapes dataset. We report FID score and IS.

|  | FID ($\downarrow$) | IS ($\uparrow$) |
|---|---|---|
| SLATE | 46.51 | 3.35 |
| SLATE+QSA | 46.23 | 3.54 |
| MOGENT (Ours) | **44.96** | **3.70** |

The figure reveals two key dynamics. First, MOGENT requires more training epochs to converge compared to SLATE. This supports that the longer training steps for MOGENT is a trade-off for its higher final performance and efficient generation. Secondly, the loss for SLATE (extended) continues to decrease throughout its training. However, the performance metrics on compositional generation task in Table 11 show that this continued decrease in loss results in large degradation in generation quality, as the FID score worsens from 46.51 to 76.12. As there is a disconnect between the loss curve and the final generation quality for SLATE, we opt to use the hyperparameters reported by the authors.

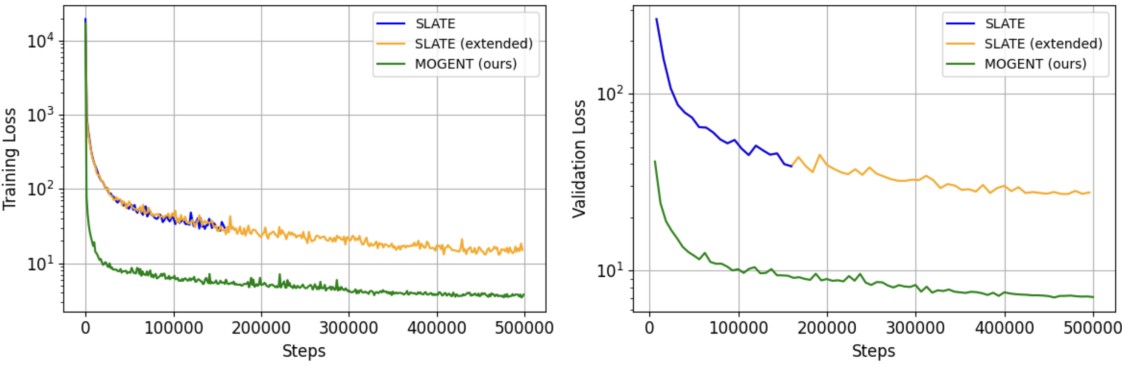

Figure 14: Visualization of training and validation losses of SLATE, SLATE trained with the same number of epochs as MOGENT (SLATE extended), and MOGENT when training on 3D Shapes dataset.

Table 11: Comparison of SLATE, SLATE trained with the same number of epochs as MOGENT (SLATE extended), and MOGENT on the compositional generation task using 3D Shapes dataset. We report FID score and IS.

|  | FID ($\downarrow$) | IS ($\uparrow$) |
| --- | --- | --- |
| SLATE | 46.51 | 3.35 |
| SLATE (extended) | 76.12 | **3.73** |
| MOGENT (Ours) | **44.96** | 3.70 |

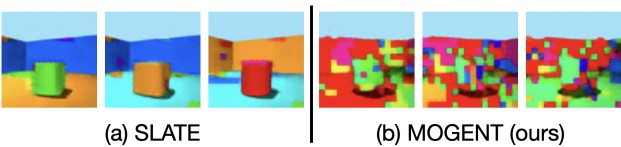

(a) SLATE      (b) MOGENT (ours)

Figure 15: Visualization of unconditional generation on 3D Shapes dataset.

### B.6 Unconditional Generation

We investigate the model's unconditional generation capabilities on the 3D Shapes dataset by replacing the object-slot conditioning with latent codes sampled from a standard Gaussian distribution. As Figure 15 shows, SLATE generates coherent compositional scenes, whereas MOGENT produces distorted, non-compositional images. Notably, while our model consistently renders the correct background color——likely because the training dataset contains only one possible sky color——the foreground objects are fragmented and non-compositional. We attribute this failure to the fundamental difference in decoding mechanisms. SLATE's autoregressive process ensures each token conditions on its predecessors, maintaining local dependencies. In contrast, the parallel decoding in MOGENT means tokens within a generation step are sampled independently of one another. Without strong slot conditioning, this lack of intra-step dependency prevents the coordinated formation of objects. This finding, which relates with a known limitation of parallel decoding (Besnier et al., 2025; Feng et al., 2025), delineates a key trade-off: our framework excels at efficient, slot-conditioned generation, but ensuring compositionality remains an important challenge, providing direction for future research.

### B.7 Ablation of SlotDiffusion

We further compare the quality-efficiency trade-off of MOGENT against SlotDiffusion with different solver hyperparameters. The official SlotDiffusion implementation adopts a 3rd-order, singlestep DPM-Solver++ (Lu et al., 2025) with 20 sampling steps. To provide a broader view, we evaluate both singlestep and multistep variants with solver orders $2, 3$. We report FID on the CLEVRTex dataset for the number of function evaluations (NFE) ranging from 1 to 20 (Figure 16).

The result reveals mainly three findings. First, the default solver choice in SlotDiffusion is not optimal: a 2nd-order singlestep solver achieves the best FID of 29.66 at NFE $= 4$, outperforming the baseline. Second, while MOGENT does not result in the lowest FID, its quality–efficiency balance is better. SlotDiffusion can reduce its NFE to as few as two steps without significant quality degradation (providing up to a $10\times$ speedup), yet MOGENT remains faster overall, reaching a $17\times$ speedup with competitive quality. Third, we observe an instability when using the 3rd-order, multistep solver. Although it yields reasonable FID for small NFEs, its performance deteriorates sharply beyond NFE $\geq 10$. This phenomenon is consistent with numerical instabilities in higher-order multistep solvers under stiff dynamics.

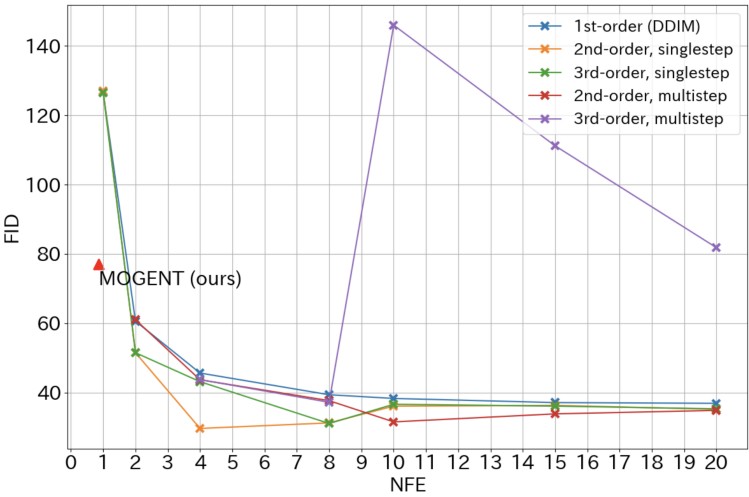

Figure 16: Ablation of SlotDiffusion with different solver hyperparameters.

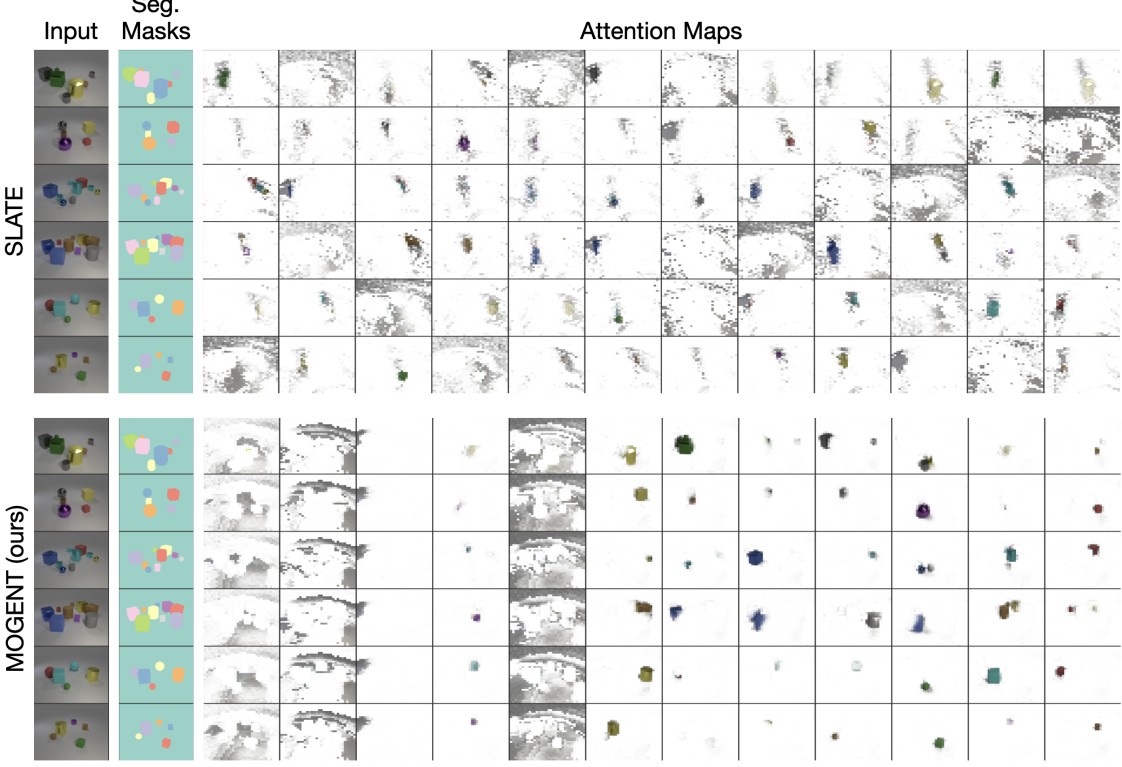

Figure 17: More visualization of attention maps of SLATE and MOGENT on CLEVR with masks dataset.

## C  Additional Qualitative Results

### C.1  Image Segmentation

We provide more visualization results of the image segmentation task (Section 4.2) on the CLEVR dataset in Figure 17. Although MOGENT fails to correctly segment objects of smaller size or similar appearances in some cases, our model attends more to individual objects compared to SLATE.

We also show visualizations of attention maps on CLEVRTex and CelebA datasets in Figure 18.

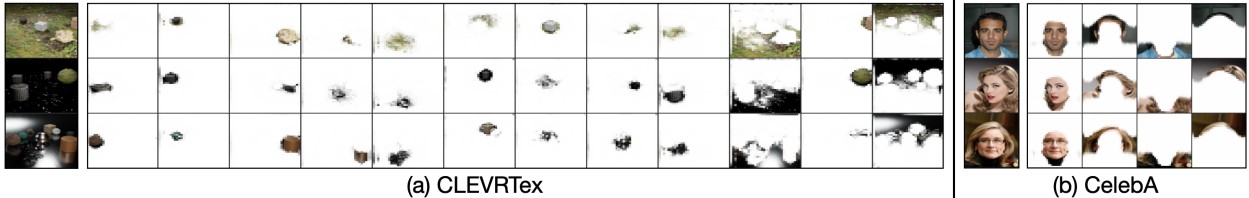

(a) CLEVRTex        (b) CelebA

Figure 18: Visualization of attention maps of MOGENT on (a) CLEVRTex and (b) CelebA datasets.

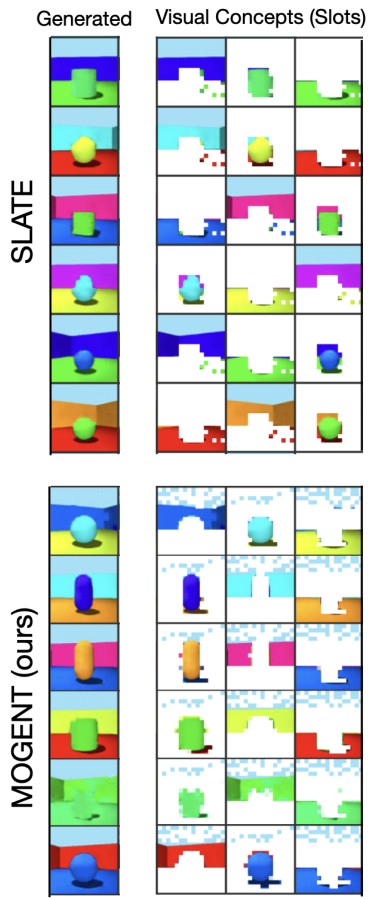

Figure 19: More visualization of generated images by SLATE and MOGENT on the 3D Shapes dataset.

## C.2    Compositional Generation

Figure 19 and Figure 20 show more visualizations from the compositional generation task (Section 3.2) on 3D Shapes and CLEVR dataset, respectively. On 3D Shapes dataset, whereas the visual concepts of SLATE look similar between samples (e.g., the attention map for the foreground object), the visual concepts of MOGENT vary more. We think this enabled MOGENT to generate images with higher fidelity. On CLEVR, we see that each visual concept attends to individual objects more, and MOGENT is able to generate images with higher fidelity and diversity.

## C.3    Image Reconstruction

Figure 21 shows more visualization results of the image reconstruction task (Appendix B.1) on the CLEVR dataset. The figure shows that while the reconstruction quality does not change much depending on the mask ratio, the model cannot reconstruct objects for which it failed to extract the corresponding slot representations. We can see that these objects tend to be smaller in size, partially occluded by other objects, or have similar colored objects nearby.

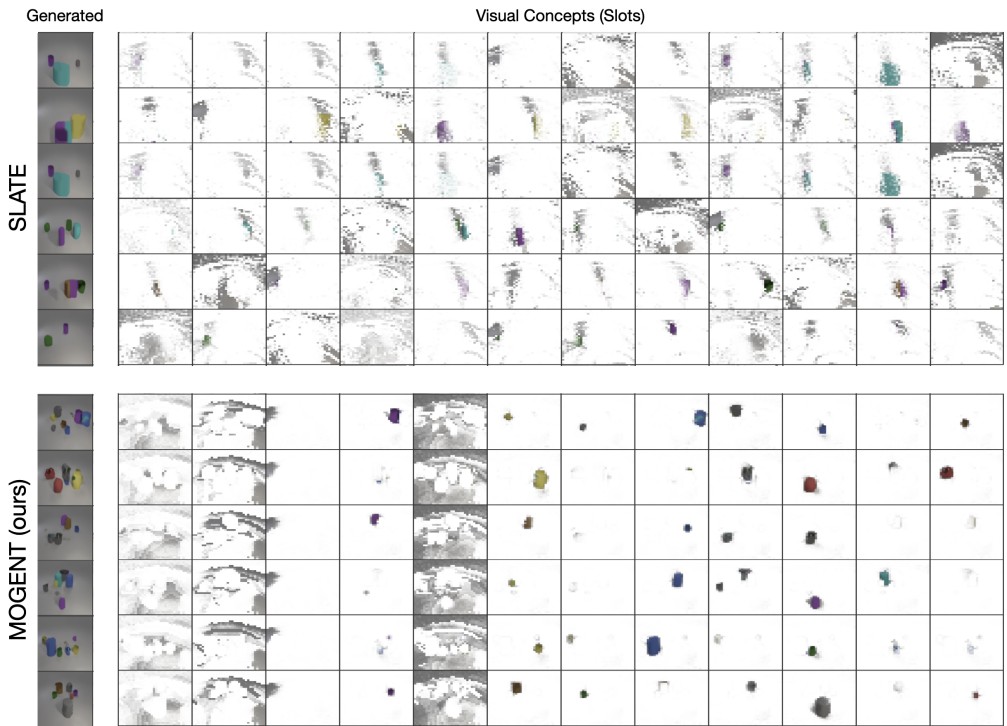

Figure 20: More visualization of generated images by SLATE and MOGENT on the CLEVR dataset.

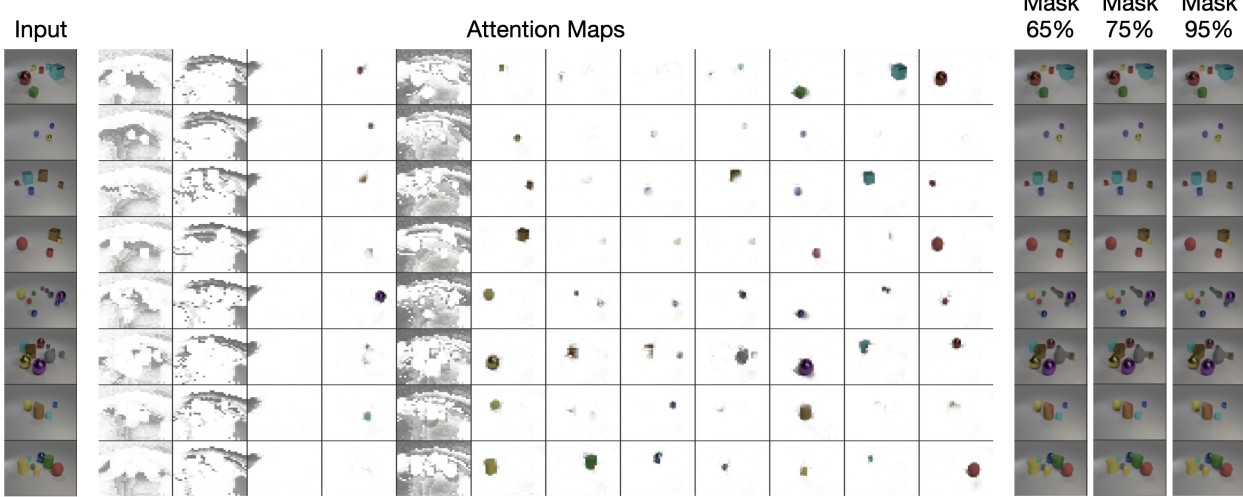

Figure 21: More visualization of attention maps and reconstructed images with different mask ratios on CLEVR with masks dataset.

## D  Extended Related Works: Accelerating Generative Models

In this section, we cover related works on accelerating the inference speed of generative models in the image domain.

**Accelerating Autoregressive Models.**  For autoregressive models, most prior works have explored utilizing non-autoregressive methods. Masked generative modeling achieves efficiency by iterative, parallel sampling (Chang et al., 2022a; Lee et al., 2022; Chang et al., 2023; Weber et al., 2024).  However, some works have pointed out that a naïve sampling scheme can be suboptimal (Besnier et al., 2025; Feng et al.,

2025; Ni et al., 2024). Semi-autoregressive methods keep an autoregressive backbone but decode blocks in parallel (Wang et al., 2025; Ren et al., 2025; Jang et al., 2025; Teng et al., 2025). Masked autoregressive models aim to unify autoregressive and masked generative models by predicting multiple tokens in a random order (Li et al., 2024c). More recently, visual autoregressive models explore hierarchical prediction strategies (Tian et al., 2024; Han et al., 2025). Orthogonal to these approaches, systems techniques such as KV caching and efficient attention implementations (e.g. FlashAttention (Dao et al., 2022; Dao, 2024; Shah et al., 2024)) can also reduce inference cost without altering the decoding rule. Our work falls into the first category.

**Accelerating Diffusion Models.** There is a line of research focusing on accelerating the inference speed of diffusion models, which traditionally require a large number of generation steps. These works can be broadly categorized into three approaches. The first approach includes architectural changes, such as using consistency models (Song et al., 2023; Luo et al., 2023; Song & Dhariwal, 2024) or shortcut models (Frans et al., 2025). Another approach is improving sampling schemes, for example, by using improved samplers (Song et al., 2021; Lu et al., 2022; 2025) or reducing the computational redundancy during inference (Li et al., 2024b; Ma et al., 2024) A third approach involves post-hoc techniques such as distillation (Luhman & Luhman, 2021; Sauer et al., 2024; Yin et al., 2024; Salimans & Ho, 2022) or parallelization across multiple GPUs (Li et al., 2024a).

MOGENT is similar to the first approach as it tackles the efficiency problem from an architectural perspective. However, in contrast to these approaches for diffusion models that shows a sharp trade-off between speed and quality, MOGENT offers a different point on the speed-quality curve. MOGENT offers better solution compared to methods that reduce computational redundancy, as these impact is most significant on samplers with many steps (e.g., 250+). Finally, since our contribution is complementary to post-hoc techniques, these methods are orthogonal to our architectural focus.

**Flow-based Models.** Parallel to diffusion, flow-based models, such as Rectified Flow (Liu et al., 2023), aim to learn near-straight transport paths that support very low-step sampling. While its effectiveness was initially examined only on smaller datasets, methods to scale it to larger datasets have been recently proposed (Liu et al., 2024; Lee et al., 2024). To our knowledge, integrating object-centric learning with flow-based models remains unexplored. We leave this direction as future work.

