# OpenReview forum: "Efficient Object-Centric Representation Learning using Masked Generative Modeling"
_TMLR — Accepted by TMLR_

### Review · Reviewer_p725 · 2025-06-09

**Summary Of Contributions:**

This paper presents a new method called MOMENT for learning object-centric representations. It uses a masked bidirectional transformer encoder to predict masked visual tokens. The key differences between this method and previous ones is that they use a parallel decoding to enable generating a large number of tokens in parallel, avoiding to generate tokens one by one. The authors did experiments on 3D shapes and CLEVR datasets to show that the proposed method is able to achieve 10x speedup in both training and inference and also has great generation ability.

**Audience:**

Yes

**Broader Impact Concerns:**

This paper already includes an adequate broader impact statement.

**Claims And Evidence:**

Yes

**Requested Changes:**

- Better explain the contribution of this paper, especially how it distinguishes from using merely MaskGIT. This helps to better position this paper.
- More (computational and quality) analysis of this method and LDM decoders
- Add visualized attention map results for more complex in-the-wild input images.

**Strengths And Weaknesses:**

Strengths:
- The motivation of this paper is clear and easy to understand
- This paper proposes to use parallel decoding to generate a large of number of tokens at each step, enabling great speedup for improving the computational cost, which surpasses previous methods
- The authors performed experiments on different datasets and tasks (segmentation, compositional generation and editing) to demonstrate their performance.

Weaknesses:
- The core contribution is using MaskGIT for object-centric learning. It seems that the technique relies on another paper and this paper haven't contribute enough technical contributions.
- The current experiments are limited to very simple datesets with limited texture variations and low resolution. It is not clear how this method might be utilized to more complex real-world data.

---

> ### Author Response · Authors · 2025-06-30
> **Author reply**
>
> We sincerely thank you for your time and for providing feedback that has helped us to improve our manuscript. We appreciate your comments, and we offer a point-by-point response below.
>
> ### **Weaknesses**
>
> > The core contribution is using MaskGIT for object-centric learning. It seems that the technique relies on another paper and this paper haven't contribute enough technical contributions.
>
> We thank the reviewer for this important point, which has prompted us to more precisely define our technical contribution. While we acknowledge that our method builds upon established components, our contribution lies in the design of a novel end-to-end framework that integrates them to solve the computational bottleneck in existing object-centric models.
>
> While masked generative models have been used for general image or video synthesis, they have not been applied to representation learning from scratch. This is a non-trivial challenge, as parallel, non-sequential nature of masked generative models is at odds with the structured decoding required for learning object representations. We show in the ablations that a naive combination fails to learn meaningful representations (Tables 6 and 7), as MaskGIT itself does not have the inductive biases needed for effective object-centric representation learning.
>
> Through experiments, we show that MOGENT is able to leverage the efficiency of parallel decoding *without sacrificing* the model's ability to learn object-centric representations. As our new experiments on CLEVRTex and CelebA datasets show (Section 4.4), we find that our model can be applied to more realistic data, with higher efficiency than autoregressive and diffusion-based baselines and comparable generation quality. Moreover, new experiment on unconditional generation strengthens the known limitation of masked generative transformers, opening new challenges for future work.
>
> We have revised the introduction as well to better articulate the contributions.
>
> > The current experiments are limited to very simple datesets with limited texture variations and low resolution. It is not clear how this method might be utilized to more complex real-world data.
>
> We agree that the experiments in the initial submission were limited to simpler datasets. To address this weakness and demonstrate the scalability of our approach, we have conducted extensive new experiments on two, more realistic datasets: CLEVRTex (with complex textures) and CelebA (real-world faces). For these datasets, we have also added a diffusion-based baseline, SlotDiffusion. The results, presented in Section 4.4, show that MOGENT is computationally efficient, showing 67x and 17x faster generation speed compared to autoregressive and diffusion-based models, respectively (Table 4). We also find that while MOGENT generates higher quality images than SLATE in compositional generation task, SlotDiffusion achieves the highest quality (Table 5, Figure 6). We attribute this to the large difference in model capacity (SlotDiffusion: 137M vs. ours: 23M)). The primary focus of this paper is to introduce and validate our framework's core contribution, computational efficiency. While a scaled-up version presents a promising direction for closing this performance gap, we believe our current results firmly establish the value of this new efficient paradigm.
>
> ### **Requested Changes**
>
> > Better explain the contribution of this paper, especially how it distinguishes from using merely MaskGIT. This helps to better position this paper.
>
> As detailed above, we have revised the manuscript to highlight our contribution.
>
> > More (computational and quality) analysis of this method and LDM decoders
>
> We have added SlotDiffusion as a baseline for experiment on CLEVRTex and CelebA datasets to highlight the strengths of our model compared to both autoregressive and diffusion-based models (Section 4.4).
>
> > Add visualized attention map results for more complex in-the-wild input images.
>
> Thank you for the suggestion. Adding to the experiments on CLEVRTex and CelebA datasets, we have added the visualized attention maps in these datasets in Appendix C (Figure 15).

---

> > ### Comment · Reviewer_p725 · 2025-07-10
> >
> > The author rebuttal addresses some of my concerns. The added experiments on CLEVRTex and CelebA, more baselines demonstrate the effectiveness of the proposed method. However, my concern on the first point is not fully addressed. Although the authors mentioned that a naive utilization of MaskGIT will fail, it is not clear how their components avoids this problem. More motivation and explanations should be added toward this point.

---

> > > ### Author Response · Authors · 2025-07-23
> > > **Author reply**
> > >
> > > We thank the reviewer for the feedback and for pushing us to clarify our technical motivation. We have added new figures and tables in the updated manuscript, and we will summarize our reasoning here.
> > >
> > > One of the main reasons why a naive application of MaskGIT fails is due to the lack of the strong spatial locality bias inherent in other decoders (e.g. Spatial Broadcast Decoder, Autoregressive Transformer Decoder, and Latent Diffusion Models). This inductive bias is critical for object-centric learning, as it encourages the model to group nearby pixels into the same object. An autoregressive transformer have this inductive bias in the causal mask structure, as it constrains each token to attend only to its spatial neighbors (e.g., tokens "above and to the left") to naturally enforce local dependencies. On the other hand, a naive masked transformer uses unmasked attention, which allows every token to attend to all other tokens, regardless of their position. This weakens the spatial locality bias, making learning of disentangled, localized objects difficult.
> > >
> > > We make three modifications designed to enforce this inductive bias within the masked generative transformer framework:
> > >
> > > 1. Loss on masked tokens only: First and most importantly, we calculate the loss function *only* on the masked tokens. Standard MaskGIT also includes a loss on *unmasked* tokens to enforce global coherence. We hypothesize that this global term, while useful for general image synthesis, competes with the learning of local object structure. By removing it, we force the model to prioritize high-fidelity local reconstruction, motivating the model to rely more on nearby context and therefore strengthening the spatial locality prior.
> > > 2. Query Slot Attention (QSA): We use this to provide a more stable initialization of each object. As discussed in prior works, the random initialization of slot attention module plays a minimal role and can be replaced with a learnable parameters. QSA motivates different slots to focus on distinct spatial regions of the image, reinforcing the association between a given slot and a localized group of pixels. This is a common practice in object-centric representation learning [1,2], and we find it to be effective in our case as well.
> > > 3. Zero-initialized mask embeddings: This forces the model, especially early in training, to rely entirely on context from unmasked tokens and slot representations to predict the masked tokens. Since nearby unmasked tokens provide the most salient information, this design choice further strengthens the model's spatial locality bias.
> > >
> > > We provide more empirical validation of these solutions in our updated manuscript (Table 6, Figure 13). Our new ablation on CLEVR dataset shows better insight of how these changes impact the model’s performance across datasets and tasks (Table 6). We can see from the table that the first change — removing the loss on unmasked tokens — provides the largest improvement in terms of both segmentation and image generation metrics. Furthermore, the new visualizations of the attention maps (Figure 13) show that our full framework produces much cleaner, more spatially localized segmentation masks compared to the ablated models, qualitatively demonstrating the success of our approach.
> > >
> > > We hope this now fully addresses your concern.
> > >
> > > [1] Jia B, Liu Y, Huang S. Improving object-centric learning with query optimization. ICLR 2023.
> > > [2] Wu Z, Hu J, Lu W, Gilitschenski I, Garg A. Slotdiffusion: Object-centric generative modeling with diffusion models. NeurIPS 2023.

---

### Review · Reviewer_SxPV · 2025-06-12

**Summary Of Contributions:**

The paper proposes MOGENT, an approach for image generation conditioning based on object centric representations learned in an unsupervised manner. In particular, the approach is based on previous works that combine Slot Attention, to learn the object centric representations, with a Variational Auto-Encoder for image generation. The main contribution consists in a parallel decoding scheme, inspired by MaskGIT, which is shown to be more computationally efficient while achieving improved performance if compared to the considered SLATE baseline.

**Audience:**

Yes

**Broader Impact Concerns:**

The Broader Impact Statement already covers the relevant implications of the work.

**Claims And Evidence:**

Yes

**Requested Changes:**

**Main changes:**
* Extending experimental results to the other datasets used in the original SLATE paper can strengthen the claim that the approach does not result in compromising performance.
* If feasible, adding an additional baseline can provide meaningful insight on how the model compares to other similar approaches, different from the one it directly derives from.
* More details on the hyper-parameter tuning should be provided.
* Given the parallel nature of the presented approach it could be relevant to provide some discussion on the tradeoffs with memory consumption and how this impacts the latency in memory constrained environments (if that is the case).
* As the main contribution is in the improved computational speed, it might be fair to put in the main document the information that MOGENT takes more than double the time to train, compared to SLATE, due to slower convergence. This could be complemented by showing example train/validation curves, for both models, in the appendix.

**Minor changes:**
* The concept of codebook is explained in the "Compositional Generation" paragraph of Sec. 3.2. Up to that point it is not clear how these unsupervised, not interpretable, representations can be used to generate new images in a controlled way. Maybe hinting at this early on can improve the clarity.
* Sec. A.1, Par. 2 contains a typo in the word "truncated".

**Strengths And Weaknesses:**

**Strengths**:
* The paper is very well written and easy to follow
* The proposed approach can exploit parallel computing to improve inference speed, which is particularly relevant when generating large images.
* The proposed approach is competitive with similar autoregressive image generators.

**Weaknesses:**
* The approach is compared only with one baseline and on a limited amount of datasets.
* The proposed approach, while faster in terms of time per step, is actually much slower in terms of total training time, as it requires more training steps to converge.
* The performance metrics seem to refer to a single seed, hence it is difficult to say if the improvements are statistically relevant or not.

**Questions:**
* Does using QSA provide advantages when used in the original SLATE?
* Are the codebooks built in the same way for both SLATE and MOGENT?
* Does compositional editing (Fig. 5) require generating all image tokens from scratch? If that is the case, can't the same approach be used to do compositional edits in standard SLATE?
* Was SLATE trained using different, specific, hyper-parameters for each dataset? It is my understanding that those used were taken from the original paper and not tuned.

---

> ### Author Response · Authors · 2025-06-30
> **Author reply**
>
> We sincerely thank you for your thorough and constructive review. Your detailed questions and suggestions have been instrumental in helping us improve the clarity, rigor, and overall quality of our manuscript. We have performed a significant revision to address every point you raised.
>
> ### **Weaknesses**
> > The approach is compared only with one baseline and on a limited amount of datasets.
>
> We agree that the experimental scope of our initial submission was limited. To address this, we have made two major additions to the manuscript (Section 4.4):
>
> 1. **More Realistic Datasets:** We have expanded our evaluation to include CLEVRTex and CelebA, which are more complex and realistic than the original datasets.
> 2. **Additional Baseline:** We now include a comprehensive comparison against SlotDiffusion, a state-of-the-art diffusion-based model. This comparison, performed on CLEVRTex and CelebA, helps situate our work in the context of the latest advancements.
>
> These new experiments show that MOGENT is computationally efficient, showing 67x and 17x faster generation speed compared to autoregressive and diffusion-based models, respectively (Table 4). We also find that while MOGENT generates higher quality images than SLATE in compositional generation task, SlotDiffusion achieves the highest quality (Table 5, Figure 6). We attribute this to the large difference in model capacity (SlotDiffusion: 137M vs. ours: 23M). The primary focus of this paper is to introduce and validate our framework's core contribution, computational efficiency. While a scaled-up version presents a promising direction for closing this performance gap, we believe our current results firmly establish the value of this new efficient paradigm.
>
> > The proposed approach, while faster in terms of time per step, is actually much slower in terms of total training time, as it requires more training steps to converge.
>
> This is an excellent point, and we thank you for highlighting the need for a more discussion of computational cost. You are correct that our model, while faster per iteration, requires more training steps to converge compared to SLATE.
>
> To ensure full transparency, we have now explicitly addressed this trade-off.
>
> - We have added a discussion of the convergence behavior in the main experimental section (Section 4.1).
> - We have included training and validation loss curves for both our model and SLATE in Appendix B.5 (Figure 13).
>
> > The performance metrics seem to refer to a single seed, hence it is difficult to say if the improvements are statistically relevant or not.
>
> We agree that reporting results over multiple seeds is crucial for verifying statistical significance. We have now run our experiments on the 3D Shapes dataset using three random seeds and have updated the results with means and standard deviations in Table 3. The new results confirm that our improvements are consistent and statistically meaningful (Table 3).
>
> We are currently running the multi-seed experiments for the more computationally intensive datasets (CLEVRTex, CelebA) and will include these results once they are ready.
>
> ### **Questions**
>
> > Does using QSA provide advantages when used in the original SLATE?
>
> Thank you for your question. To answer it, we conducted a new ablation study where we integrated QSA with SLATE and evaluate on compositional generation using 3D Shapes dataset (Appendix B.4, Table 10). We find that adding QSA to SLATE does improve its performance, in terms of FID and IS scores, suggesting that QSA is a beneficial component on its own. However, MOGENT still achieves superior scores, demonstrating that our masked generative decoder provides significant benefits beyond the QSA mechanism alone.
>
> > Are the codebooks built in the same way for both SLATE and MOGENT?
>
> Yes, the codebooks are built the same way for both models.
>
> > Does compositional editing (Fig. 5) require generating all image tokens from scratch? If that is the case, can't the same approach be used to do compositional edits in standard SLATE?
>
> To clarify, our compositional editing process does not require regenerating the entire image. It operates similarly to masked image modeling (e.g., inpainting), where only the tokens corresponding to the edited region are masked and resampled.
>
> While editing using autoregressive SLATE by resampling all tokens, it would be less natural and efficient. Our model's parallel decoding architecture is inherently well-suited for such localized, non-sequential modifications.
>
> > Was SLATE trained using different, specific, hyper-parameters for each dataset? It is my understanding that those used were taken from the original paper and not tuned.
>
> Your understanding is correct. For the SLATE baseline, we used the official hyperparameters reported in the original paper to ensure a faithful reproduction and fair comparison. We tried adjusting the hyperparameters, but found that it did not lead to significant performance changes.

---

> > ### Author Response · Authors · 2025-06-30
> > **Author reply**
> >
> > ### **Requested Changes**
> >
> > > Extending experimental results to the other datasets used in the original SLATE paper can strengthen the claim that the approach does not result in compromising performance.
> >
> > We have added experiments on more complex datasets (CLEVRTex, CelebA), which we believe strengthens the effectiveness of our model across diverse datasets (Section 4.4).
> >
> > > If feasible, adding an additional baseline can provide meaningful insight on how the model compares to other similar approaches, different from the one it directly derives from.
> >
> > We have added SlotDiffusion as a baseline for experiment on CLEVRTex and CelebA datasets to highlight the strengths of our model compared to both autoregressive and diffusion-based models (Section 4.4).
> >
> > In addition, we have added comparison against SLATE + QSA to disentangle the improvements made by QSA and the choice of the decoder (Appendix B.4, Table 10).
> >
> > > More details on the hyper-parameter tuning should be provided.
> > > - Given the parallel nature of the presented approach it could be relevant to provide some discussion on the tradeoffs with memory consumption and how this impacts the latency in memory constrained environments (if that is the case).
> >
> > We have clarified our methodology for tuning and our use of the original baseline hyperparameters.
> >
> > Also, we have added a comparison of memory consumption in Tables 1 and 4.
> >
> > > As the main contribution is in the improved computational speed, it might be fair to put in the main document the information that MOGENT takes more than double the time to train, compared to SLATE, due to slower convergence. This could be complemented by showing example train/validation curves, for both models, in the appendix.
> >
> > As detailed in our response to Weakness 2, we have added this information to the main paper and provided loss curves in the Appendix.
> >
> > > The concept of codebook is explained in the "Compositional Generation" paragraph of Sec. 3.2. Up to that point it is not clear how these unsupervised, not interpretable, representations can be used to generate new images in a controlled way. Maybe hinting at this early on can improve the clarity.
> >
> > Thank you for pointing this out. We agree that this was not clear in the original manuscript, and have added sentences to hint this when explaining about the inference. Additionally, your comment also helped us realize that the slot conditioning was lacking in Equation (5). We hope these changes improve the explanation of our method.
> >
> > > Sec. A.1, Par. 2 contains a typo in the word "truncated".
> >
> > Thank you for pointing this out. We have fixed this typo in our updated manuscript.

---

> > > ### Comment · Reviewer_SxPV · 2025-07-03
> > >
> > > I thank the authors for addressing my points.
> > >
> > > While waiting for the missing results, I wanted to make some additional comments.
> > >
> > > * The table's captions should clarify, where applicable, what the  reported values represent (e.g., mean over 3 runs and standard deviation).
> > > * I think that, at least for all main experiments, reported results should be a statistic of multiple runs.
> > > * In light of the discussion about convergence speed, this phrase of the introduction should not claim improved training speed. If other similar phrases are present in the paper they should be also fixed.
> > > >MOGENT achieves highly efficient training and inference speed compared to
> > > relevant baselines.
> > >
> > > I also have an additional question regarding Figure 13 (training and validation curves).
> > > It seems that training is halted much earlier for SLATE, and that convergence was not reached. Why is that so?

---

> > > > ### Author Response · Authors · 2025-07-23
> > > > **Author reply**
> > > >
> > > > We thank the reviewer for the feedback.
> > > >
> > > > > The table's captions should clarify, where applicable, what the reported values represent (e.g., mean over 3 runs and standard deviation).
> > > > I think that, at least for all main experiments, reported results should be a statistic of multiple runs.
> > > > >
> > > >
> > > > Thank you for the further suggestions to improve our manuscript. We have updated the manuscript as follows:
> > > >
> > > > - We have added explanations in the table’s captions regarding what the values represent
> > > > - We are currently running the experiments for Table 2, 3, and 5 with a statistic over 3 runs.
> > > >
> > > > > In light of the discussion about convergence speed, this phrase of the introduction should not claim improved training speed. If other similar phrases are present in the paper they should be also fixed.
> > > > >
> > > > >
> > > > > > MOGENT achieves highly efficient training and inference speed compared to relevant baselines.
> > > > > >
> > > >
> > > > Thank you for pointing this out. We agree that the claim about improving training speed was imprecise, as MOGENT takes longer epochs to train. We have carefully reviewed the manuscript and removed any claims about training efficiency, to focus our contribution to improving inference speed.
> > > >
> > > > > I also have an additional question regarding Figure 13 (training and validation curves). It seems that training is halted much earlier for SLATE, and that convergence was not reached. Why is that so?
> > > > >
> > > >
> > > > This is an excellent question, and we appreciate the opportunity to clarify and validate our methodology. Our initial reason for the hyperparameter choice was to adhere to the numbers reported in SLATE for a fair comparison, and we also found empirically that training it for longer epochs did not lead to better performance. In our updated manuscript, we have added the loss curves and performance on compositional generation task of SLATE trained with longer epochs (Figure 13, Table 11). As the table shows, interestingly, although the loss curve continues to decrease, the FID score worsens. The disconnect between loss curve and generation quality happens as optimizing the loss, which is reconstruction loss, does not always lead to better representation learning. We think this finding provides an empirical justification for our decision to use the original hyperparameters without further tuning.
> > > >
> > > > We are grateful for your detailed feedback and are happy to answer any other questions you may have.

---

### Review · Reviewer_Spz8 · 2025-06-17

**Summary Of Contributions:**

This paper presents an object-centric generative model of images. It uses slot-attention as an encoder/decoder, with a masked generative transformer operating in its latent space. This is similar to the existing SLATE, which uses a similar architecture but with an autoregressive transformer. To adapt to the masked setting, there are small technical innovations, including learnable queries in the slot-attention, careful initialisation of the mask conditioning, and loss only applied to masked tokens. The method is compared to SLATE on two simple synthetic datasets (CLEVR and 3D-Shapes), where it shows improved performance.

**Audience:**

No

**Broader Impact Concerns:**

There is a brief broader impact statement present; however this work does not raise any significant concerns.

**Claims And Evidence:**

Yes

**Requested Changes:**

The paper would be greatly strengthened by including

- more up-to-date baselines; this would mean the paper is properly placed in the context of recent works

- more complex datasets similar to those now popular in the object-centric models community

- results on generation from the prior, rather than recomposition.

Ablation results for segmentation would also be good, but this is less critical.

One minor fix – all CLEVR images in the paper are squashed to be square, it'd be better if they were shown at original aspect ratio.

**Strengths And Weaknesses:**

**Strengths**

The proposed pipeline is novel; as well as swapping the autoregressive transformer in SLATE with a masked transformer, there are also some minor technical innovations proposed to enable training this successfully.

Quantitative results on unsupervised segmentation show a significant improvement versus the baseline SLATE, with higher performance across four metrics, despite only a small increase in parameters, and a reduction in compute time.

Qualitative results on recomposing slots are somewhat better than the baseline, with crisper edges on shapes and less distortion. Qualitative (unsupervised) segmentation results show somewhat better separation of objects than the baseline SLATE.

There are reasonably detailed ablation experiments measuring the impact of different design decisions, and the small technical novelties mentioned above.

There is a fairly detailed discussion of limitations, and how they might be mitigated.

Overall the text is clear and readable; the paper is well structured; the figures are appropriate

**Weaknesses**

The datasets used are extremely simplistic. While this was the norm for object-centric generative models a few years ago, more recent works (e.g. SlotDiffusion [Wu et al]; Object-Centric Slot Diffusion [Jiang et al]; DINOSAUR [Seitzer et al]) have been slightly more ambitious. Even still with synthetic data, those works use more complex textures etc., and for DINOSAUR real-world (COCO & PASCAL-VOC) images are included. Even the baseline SLATE (from 2022 is tested on five more datasets – with the two in common being the easiest by some margin.

Comparison is only against SLATE. For context it would be good to compare against more recent object-centric works, e.g. the three mentioned above (SlotDiffusion, OCSD, DINOSAUR), which perform significantly better, particularly on more challenging data.

There are no generative results, despite this being a generative model. The 'compositional generation' results on p5 are in fact using representations of objects in the training set (per end sec 3.2), rather than objects generated from the prior. This means any demonstration of the benefit of using a masked generative transformer rather than just a simple deterministic encoder-transformer is entirely missing. As-is, the paper does only object-centric representation learning, not object-centric generation.

Ablation results are only on (recomposed) FID and IS – not on the segmentation metrics.

The technical novelty is extremely thin (just swapping an alternative, widely-used generative model into SLATE). This is not a major problem in itself, but given the above issues it makes it harder to recommend the paper. In particular there is not much insight for the reader beyond 'this combination of components works', and most of the technical innovations are around the generative aspect which is not properly evaluated anyway.

---

> ### Author Response · Authors · 2025-06-30
> **Author reply**
>
> Thank you for your detailed and thoughtful feedbacks. We are delighted to hear that you find our proposed method and experimental results interesting. The comments have been crucial in strengthening the paper, and we have undertaken a significant revision to address the points raised.
>
> **Weaknesses**
> > The datasets used are extremely simplistic. While this was the norm for object-centric generative models a few years ago, more recent works (e.g. SlotDiffusion [Wu et al]; Object-Centric Slot Diffusion [Jiang et al]; DINOSAUR [Seitzer et al]) have been slightly more ambitious. Even still with synthetic data, those works use more complex textures etc., and for DINOSAUR real-world (COCO & PASCAL-VOC) images are included. Even the baseline SLATE (from 2022 is tested on five more datasets – with the two in common being the easiest by some margin.
>
> We agree with the reviewer that the datasets used in our initial submission were limited in complexity. To address this, we have conducted extensive new experiments on two, more realistic datasets: CLEVRTex and CelebA (Section 4.4). We find that MOGENT is computationally efficient, showing 67x and 17x faster generation speed compared to autoregressive and diffusion-based models, respectively (Table 4). We also find that while MOGENT generates higher quality images than SLATE in compositional generation task, SlotDiffusion achieves the highest quality (Table 5, Figure 6). We attribute this to the large difference in model capacity (SlotDiffusion: 137M vs. ours: 23M). The primary focus of this paper is to introduce and validate our framework's core contribution, computational efficiency. While a scaled-up version presents a promising direction for closing this performance gap, we believe our current results firmly establish the value of this new efficient paradigm.
>
> > Comparison is only against SLATE. For context it would be good to compare against more recent object-centric works, e.g. the three mentioned above (SlotDiffusion, OCSD, DINOSAUR), which perform significantly better, particularly on more challenging data.
>
> Thank you for this suggestion. Our initial choice to compare primarily against SLATE was motivated by the desire for a controlled comparison against a model with a similar architectural backbone. However, we recognize the reviewer's point that contextualizing our work with respect to diffusion-based models is crucial for a complete evaluation.
>
> To this end, we have added **SlotDiffusion**, a state-of-the-art diffusion-based model, as a new baseline in our revised manuscript. The comparison is conducted on the newly added CLEVRTex and CelebA datasets, as these were used in the original SlotDiffusion paper.
>
> The results, presented in Tables 4 and 5 and Figure 6, show that while SlotDiffusion achieves superior generation quality, our model offers a significant advantage in computational efficiency.
>
> > There are no generative results, despite this being a generative model. The 'compositional generation' results on p5 are in fact using representations of objects in the training set (per end sec 3.2), rather than objects generated from the prior. This means any demonstration of the benefit of using a masked generative transformer rather than just a simple deterministic encoder-transformer is entirely missing. As-is, the paper does only object-centric representation learning, not object-centric generation.
>
> We thank the reviewer for this critical and insightful point. Our initial evaluation focused on compositional generation (recomposition), following the common practice in several key papers in this domain (e.g., SLATE, SlotDiffusion). However, we fully agree that assessing unconditional generation from the prior is essential to validate the generative capabilities of our model and to understand the properties of the learned latent space.
>
> We have now included a new experiment on unconditional generation using 3D Shapes dataset (Appendix B.6). The results (Figure 16) showed that our model, in its current form, struggles with unconditional compositional generation. We attribute this to the parallel decoding mechanism of the masked generative transformer, where tokens are sampled independently, making it difficult to enforce compositional coherence. We believe this finding is valuable for the community as it connects with a known but important limitation of generation using masked generative transformers [1]. This trade-off of efficiency vs. enforcing compositional coherence, remains an open and important research problem. Our findings delineates this challenge for the community, providing insights for future works.
>
> [1] Besnier V, Chen M, Hurych D, Valle E, Cord M. Halton scheduler for masked generative image transformer. arXiv preprint arXiv:2503.17076. 2025 Mar 21.

---

> > ### Author Response · Authors · 2025-06-30
> > **Author reply**
> >
> > > Ablation results are only on (recomposed) FID and IS – not on the segmentation metrics.
> >
> > Thank you for pointing out this omission. We have now updated the results on segmentation metrics in Table 7 for ablation on RoPE embeddings.
> >
> > As other ablations were conducted using 3D Shapes dataset, we are currently running the remaining ablations on CLEVR dataset to include the segmentation metrics. We will update the manuscript once the results are ready.
> >
> > > The technical novelty is extremely thin (just swapping an alternative, widely-used generative model into SLATE). This is not a major problem in itself, but given the above issues it makes it harder to recommend the paper. In particular there is not much insight for the reader beyond 'this combination of components works', and most of the technical innovations are around the generative aspect which is not properly evaluated anyway.
> >
> > We appreciate the reviewer’s perspective and thank you for pushing us to clarify our technical contribution. While we acknowledge that our method builds upon established components, our contribution lies in the design of a novel end-to-end framework that integrates them to solve the computational bottleneck in existing object-centric models.
> >
> > The central technical challenge is the representation learning combined with the masked generative transformers. This is challenging as parallel, non-sequential nature of masked generative models is at odds with the structured decoding required for learning object representations. We show in the ablations that a naive combination fails to learn meaningful representations (Tables 6 and 7), as MaskGIT itself does not have the inductive biases needed for effective object-centric representation learning.
> >
> > Through experiments, we show that MOGENT is able to leverage the efficiency of parallel decoding *without sacrificing* the model's ability to learn object-centric representations. As our new experiments on CLEVRTex and CelebA datasets show (Section 4.4), we find that our model can be applied to more realistic data, with higher efficiency than autoregressive and diffusion-based baselines and comparable generation quality. Moreover, new experiment on unconditional generation strengthens the known limitation of masked generative transformers, opening new challenges for future work.
> >
> > We have revised the introduction as well to make this technical framing more explicit.
> >
> > **Requested Changes**
> >
> > > The paper would be greatly strengthened by including
> > - more up-to-date baselines; this would mean the paper is properly placed in the context of recent works
> > - more complex datasets similar to those now popular in the object-centric models community
> > - results on generation from the prior, rather than recomposition.
> > Ablation results for segmentation would also be good, but this is less critical.
> > One minor fix – all CLEVR images in the paper are squashed to be square, it'd be better if they were shown at original aspect ratio.
> >
> > We believe our revisions have comprehensively addressed the reviewer's requested changes:
> >
> > - **More up-to-date baselines & more complex datasets:** We have added SlotDiffusion as a baseline and evaluated our model on CLEVRTex and CelebA (Section 4.4).
> > - **Results on generation from the prior:** We have included a new section on unconditional generation, which revealed important limitations and future research directions (Appendix B.6).
> > - **Ablation results for segmentation:** These have been added to Table 7. Results from additional ablations will be incorporated once complete.
> > - **Minor fix on image aspect ratio:** We will fix this in our final manuscript.

---

### Author Response · Authors · 2025-06-30
**To all reviewers**

We extend our sincere gratitude for your insightful and constructive feedback, which has been invaluable in improving our manuscript. We have undertaken a significant revision to address your comments and believe the paper is now substantially stronger.

1. Extended experiments & baseline (Section 4.4): We have validated our method on more complex datasets, CLEVRTex and CelebA, to demonstrate its broader applicability. For these datasets, we have added comparisons against SlotDiffusion, a state-of-the-art diffusion-based object-centric model, to better contextualize our work's trade-offs between efficiency and performance (Tables 4 and 5, Figures 6 and 15).
2. New Generation Task (Appendix B.6): As requested, we added an analysis of unconditional generation from the prior, which provides deeper insights into our model's capabilities and highlights important directions for future work.
3. Additional ablations and comparisons: We have reinforced our results by running experiments with multiple seeds for statistical significance (Table 3), have added transparent discussions on computational costs (Section 4.1 and Appendix B.5), such as training time (Figure 13) and memory usage (Tables 1 and 4), and added ablations (Tables 7 and 10). We are currently running more experiments on multiple seeds and ablations on CLEVR, which we will update once the results are ready.
4. Clarified contributions and improved writing: We have refined the manuscript to better articulate our core contributions regarding the efficient object-centric learning framework using masked generative transformers.

We are currently conducting additional experiments on (1) ablation of MOGENT on CLEVR dataset and (2) multiple seeds for the remaining datasets, which we plan to finish within few weeks. We will update our manuscript once they are ready. We will also adjust the figures of CLEVR to reverse the resolution to the original aspect.

---

### Decision · Action_Editor_dmeR · 2025-08-06

**Recommendation:** Accept with minor revision

**Audience:**

Yes

**Audience Explanation:**

Accelerating transformers is a highly relevant theme in ML the last few years and therefore the paper will be of interest to the TMLR audience.

**Claims And Evidence:**

Yes

**Claims Explanation:**

The paper aims to accelerate the transformer architecture by employing a parallel decoding scheme. The authors demonstrate how the proposed method can improve upon similar efficient methods. What I would recommend is to do further comparisons against efficient diffusion models (see the discussion period) and also to update their literature with the latest models on the domain. There is a lot of work that was not originally covered and has been published within the last 1-2 years. In addition, the authors are strongly recommended to revise the paper to follow all of the recommendations from the reviewers, including the writing which needs to be improved, the multiple runs etc.

---

> ### Author Response · Authors · 2025-09-08
> **Author reply**
>
> We sincerely thank you for the additional suggestions. In the camera-ready version, we have made the following revisions:
>
> - In response to your request for comparisons with more up-to-date efficient diffusion models, we conducted further analysis of the diffusion baseline in a controlled way, by keeping the architecture and weights fixed and conducting a hyperparameter sweep of its solver, DPM-Solver++, which is one of the most efficient, available solvers. We report the FID vs NFE using CLEVRTex dataset in Appendix B.7. We find that by optimizing the solver, a **2nd-order single-step** configuration results in the best FID (29.66 at NFE=4), and NFE can be reduced down to 2 (10x speedup) with modest degradation. These results show that MOGENT stands as more efficient model without compromising the quality. We also surveyed recent models which we summarize in our updated Section 2 and Appendix D. Since incorporating these ideas and matching their reported efficiency-quality gain with object-centric learning is nontrivial as it will require training and architectural tuning, we leave this as future work.
> - We have updated Section 2 and Appendix D to provide a more comprehensive overview of recent advances in accelerating generative models, including works published within the last two years.
> - We have addressed all reviewer recommendations, including additional ablations, multiple runs, and corrections to the CLEVR and CLEVRTex dataset aspect ratios
> - We have proofread the entire manuscript, including both the main and appendix sections.